# The native ORAI channel trio underlies the diversity of Ca$^{2+}$ signaling events

Ryan E. Yoast[1,8], Scott M. Emrich[1,8], Xuexin Zhang[1], Ping Xin[1], Martin T. Johnson[1], Adam J. Fike [1], Vonn Walter [2,3,4], Nadine Hempel [4,5], David I. Yule[6], James Sneyd[7], Donald L. Gill [1] & Mohamed Trebak [1,4✉]

The essential role of ORAI1 channels in receptor-evoked Ca$^{2+}$ signaling is well understood, yet little is known about the physiological activation of the ORAI channel trio natively expressed in all cells. The roles of ORAI2 and ORAI3 have remained obscure. We show that ORAI2 and ORAI3 channels play a critical role in mediating the regenerative Ca$^{2+}$ oscillations induced by physiological receptor activation, yet ORAI1 is dispensable in generation of oscillations. We reveal that ORAI2 and ORAI3 channels multimerize with ORAI1 to expand the range of sensitivity of receptor-activated Ca$^{2+}$ signals, reflecting their enhanced basal STIM1-binding and heightened Ca$^{2+}$-dependent inactivation. This broadened bandwidth of Ca$^{2+}$ influx is translated by cells into differential activation of NFAT1 and NFAT4 isoforms. Our results uncover a long-sought role for ORAI2 and ORAI3, revealing an intricate control mechanism whereby heteromerization of ORAI channels mediates graded Ca$^{2+}$ signals that extend the agonist-sensitivity to fine-tune transcriptional control.

[1] Department of Cellular and Molecular Physiology, The Pennsylvania State University College of Medicine, 500 University Drive, Hershey, PA 17033, USA. [2] Department of Public Health Sciences, The Pennsylvania State University College of Medicine, 500 University Drive, Hershey, PA 17033, USA. [3] Department of Biochemistry and Molecular Biology, The Pennsylvania State University College of Medicine, 500 University Drive, Hershey, PA 17033, USA. [4] Penn State Cancer Institute and The Pennsylvania State University College of Medicine, 500 University Drive, Hershey, PA 17033, USA. [5] Department of Pharmacology, The Pennsylvania State University College of Medicine, 500 University Drive, Hershey, PA 17033, USA. [6] Department of Pharmacology and Physiology, University of Rochester Medical Center School of Medicine and Dentistry, 601 Elmwood Avenue, Box 711, Rochester, NY 14642, USA. [7] Department of Mathematics, The University of Auckland, 38 Princes Street, Auckland 1010, New Zealand. [8] These authors contributed equally: Ryan E. Yoast, Scott M. Emrich. ✉email: mtrebak@psu.edu

Cytosolic calcium ($Ca^{2+}$) signals are tightly regulated and result from the coordinated crosstalk between multiple channels in the plasma membrane (PM) and internal organelles[1–3]. In non-excitable cells, stimulation of various PM receptors leads to the production of the diffusible second messenger inositol 1,4,5-trisphosphate ($IP_3$)[4]. $IP_3$ binds to the $IP_3$ receptor ($IP_3R$) channels[4,5] in the endoplasmic reticulum (ER) and induces the release of ER luminal $Ca^{2+}$ into the cytosol, resulting in ER store depletion. It is clearly established that ER store depletion causes a conformational change in the $Ca^{2+}$ sensing protein, stromal interacting molecule (STIM), which translocate to ER–PM junctions to trap and activate ORAI1 channels and induce store-operated $Ca^{2+}$ entry (SOCE)[6–10]. When ectopically co-expressed with STIM1, the ORAI1 homologs ORAI2 and ORAI3 can clearly mediate SOCE and its biophysical manifestation, the $Ca^{2+}$ release-activated $Ca^{2+}$ current ($I_{CRAC}$)[11,12]. However, the contribution of ORAI2 and ORAI3 to native SOCE and $I_{CRAC}$ remains uncertain.

To reliably measure SOCE function with fluorescent dyes, most studies have relied on protocols that cause maximal SOCE activation. This is achieved by the use of either pharmacological tools such as thapsigargin, which is a sarcoplasmic/ER $Ca^{2+}$ ATPase (SERCA) blocker or high agonist concentrations ($Ag^{High}$; 30–100 μM carbachol (Cch) in HEK293 cells), both of which maximally deplete ER $Ca^{2+}$ stores[13]. Under these conditions of maximal store depletion, the $Ca^{2+}$ signal typically generated is dominated by a sustained cytosolic $Ca^{2+}$ plateau phase[14]. In reality, such monotonic $Ca^{2+}$ signals represent only a narrow window of the repertoire of $Ca^{2+}$ signals stimulated by physiological receptor stimulation. In most in vivo settings, cells respond to much lower concentrations of tissue and circulating agonists[15]. In fact, at these low range ($Ag^{Low}$; 1–10 μM Cch) or mid-range ($Ag^{Mid}$; 10–30 μM Cch) concentrations, $Ca^{2+}$ signals in most cells are typically shaped as transient regenerative oscillations or spikes. The frequency of these $Ca^{2+}$ oscillations increase with increasing agonist concentrations until they "tetanize" into plateaus at high agonist concentrations[4,16–20].

Although $Ca^{2+}$ oscillations in most cells, but not all, are the direct result of $Ca^{2+}$ release through $IP_3R$, they last only for one to three spikes before running down when extracellular $Ca^{2+}$ is omitted or when SOCE is inhibited[14,19], highlighting the crucial role of $Ca^{2+}$ entry in sustaining long-term $Ca^{2+}$ oscillations and gene transcription. It is thought that specific frequency signatures of $Ca^{2+}$ oscillations are decoded by downstream effector proteins to selectively activate specific isoforms of transcription factors, including those of nuclear factor of activated T-cells (NFAT). Furthermore, despite the fact that $Ca^{2+}$ oscillations originate from ER release, there is compelling evidence that the activation of certain gene programs, specifically NFAT-dependent genes, is controlled by $Ca^{2+}$ trickling from ORAI channels[14,21]. Thus, spatially restricted $Ca^{2+}$ signaling microdomains at the mouth of ORAI channels exert a crucial control over cellular physiology.

To determine the signaling functions of native ORAI3, ORAI2, and ORAI1, we generated multiple single, double, and triple ORAI isoform knockouts in HEK293 cells using CRISPR/Cas9 technology. Using this system, we show that although ORAI1 is required for SOCE activated by maximal store depletion, surprisingly it is dispensable for $Ca^{2+}$ oscillations. We show that native ORAI3 and ORAI2 are sufficient to sustain regenerative $Ca^{2+}$ oscillations but do not significantly contribute to $Ca^{2+}$ plateaus, which are predominantly supported by ORAI1. This ORAI isoform-dependent behavior is determined by the magnitude of each isoform interaction with STIM1 under unstimulated conditions matched by the strength of fast $Ca^{2+}$-dependent inactivation (CDI) of each isoform, with ORAI3 being the highest, ORAI2 the intermediate, and ORAI1 the lowest. In the absence of ORAI2 and ORAI3, the physiological $Ca^{2+}$ responses to low- and mid-range agonist concentrations manifest mostly as plateaus, effectively shrinking the bandwidth of agonist-evoked $Ca^{2+}$ signals. We also show that both ORAI2 and ORAI3 negatively regulate ORAI1 by enhancing the channel CDI through heteromerization. Our data suggest that the ORAI isoform trio functions as a heteromultimer to form the native CRAC channel. Furthermore, we show that $Ca^{2+}$ entry through ORAI1 alone is necessary and sufficient to drive nuclear translocation of both NFAT1 (which requires robust $Ca^{2+}$ to activate) and NFAT4 (which requires modest $Ca^{2+}$ to activate), whereas ORAI2 and ORAI3 either alone or together do not support NFAT1 and NFAT4 nuclear translocation. However, ORAI2 and ORAI3 negatively regulate ORAI1-dependent NFAT1/4 nuclear translocation, in further support of native ORAI isoform heteromerization. The major consequence of ORAI2- and ORAI3-negative regulation of CRAC channels is the differential nuclear translocation of NFAT1 and NFAT4 in response to distinct agonist strengths and cytosolic $Ca^{2+}$ concentrations. Our findings identify the signaling functions of ORAI2 and ORAI3. Native ORAI2 and ORAI3 heteromerize with ORAI1 to form the native CRAC channel and are required for the graded diversity of $Ca^{2+}$ signaling events covering the full range of agonist concentrations/strengths. This broadened bandwidth of $Ca^{2+}$ signals ensures the precise nuclear translocation of NFAT isoforms of transcription factors in response to their specific range of agonist strengths and cytosolic $Ca^{2+}$ concentrations.

## Results

**ORAI1 is dispensable for regenerative $Ca^{2+}$ oscillations.** Using CRISPR/Cas9 technology, we generated clones of HEK293 cells lacking either one, two, or three ORAI isoforms. Currently, there are no reliable antibodies against ORAI2 and ORAI3. Thus, we used two guide RNAs (gRNAs) to remove a large portion of the ORAI gene. Hence, the absence of the specific mRNA is a reliable means for documenting knockout. To rule out off-target effects, we generated several independent clones for each specific ORAI isoform. For each condition shown hereafter, we show data from these two independent clones. ORAI1 protein-knockout was documented with genomic sequencing and by western blotting (Supplementary Fig. 1a, c, see also ref. [21] and Supplementary Table 4). Due to lack of reliable antibodies for ORAI2 and ORAI3, knockout of these genes was documented by genomic sequencing, PCR on genomic DNA, and at the mRNA level using quantitative reverse transcriptase-PCR (Supplementary Fig. 2 and Supplementary Table 4). It was equally important to demonstrate lack of compensatory changes in protein expression of the major players in SOCE and $Ca^{2+}$ oscillations, including both STIM isoforms (Supplementary Fig. 1b, d, e) and the three isoforms of the $IP_3R$[5,22] (Supplementary Fig. 1f–i). Furthermore, there was no significant compensatory changes in ORAI1, ORAI2, or ORAI3 mRNA as a consequence of single or double ORAI knockout (Supplementary Table 5).

Using different agonist concentrations, we empirically established three unique ranges of the muscarinic receptor agonist Cch defined as follows: (1) $Ag^{Low}$, 1–10 μM Cch; (2) $Ag^{Mid}$, 10–30 μM Cch; and (3) $Ag^{High}$, 30–100 μM Cch. We first stimulated wild-type (WT) HEK293 cells with 10 μM Cch to elicit $Ca^{2+}$ oscillations in the presence of 2 mM extracellular $Ca^{2+}$. Traces from five representative cells are shown in Fig. 1a. An average of 65% of WT HEK293 cells stimulated with 10 μM Cch responded with sustained regenerative $Ca^{2+}$ oscillations that lasted for the duration of the recording, with an average of 6 oscillations/14 min (Fig. 1f). Of the cells that did not oscillate, 26% responded with a sustained plateau, whereas the remaining 9% of cells did not respond (Fig. 1n–p).

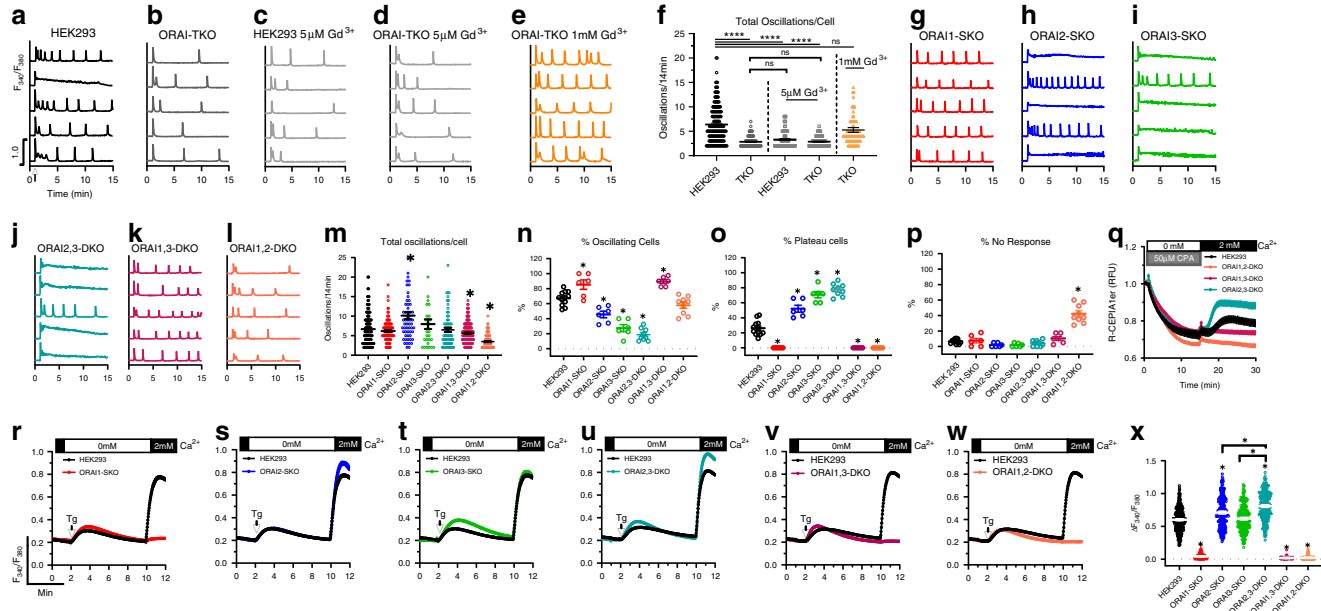

**Fig. 1 SOCE but not ORAI1 is required for maintenance of Ca²⁺ oscillations. a–e** Representative Ca²⁺ oscillations in response to 10 μM carbachol (Cch) measured using Fura2 in wild-type HEK293 and ORAI-TKO cells. Cells were maintained in HBSS containing 2 mM Ca²⁺ and stimulated with 10 μM Cch at 1 min (indicated by arrow in "a"). Representative traces from five cells/condition were chosen to represent the datasets as a whole. In **c, d**, cells were treated with 5 μM Gd³⁺ to block SOCE. In **e**, ORAI-TKO cells were treated with 1 mM Gd³⁺ (so-called Gd³⁺ insulation to block both SOCE and Ca²⁺ extrusion). **f** Quantification of total oscillations/14 min for all conditions from **a–e** (from left to right $n = 80, 84, 180, 127,$ and $78$; $n$-values correspond to individual cells). **g–l** Representative Ca²⁺ oscillations in response to 10 μM Cch measured using Fura2 over the course of 15 min in ORAI single (SKO) and double (DKO) knockout cell clones. **m–p** Quantification of total oscillations/14 min (**m**) (from left to right $n = 190, 111, 58, 29, 206, 214,$ and $67$; $n$-values correspond to individual cells), % of oscillating cells (**n**), % of plateau cells (o), and % of non-responding cells (**p**) for data in **g–l** (for **n–p**, from left to right $n = 12, 6, 6, 6, 9, 6,$ and $9$; $n$-values correspond to independent experiments). **q** Direct ER Ca²⁺ measurements using CEPIA1er in parental HEK293 cells and ORAI-DKO cells. **r–w** Measurements of ER Ca²⁺ release (in 0 mM Ca²⁺) and SOCE (in 2 mM Ca²⁺) on store depletion with 2 μM thapsigargin (Tg) in parental HEK293 cells and ORAI-SKO and DKO cells; all traces are plotted as mean ± SEM. **x** Quantification of SOCE magnitude for all conditions from **r–w**. Scatter plots (**f, m, x**) are represented as mean ± SEM and were statistically analyzed using a Kruskal–Wallis one-way ANOVA with multiple comparisons to WT HEK293 where (*$p < 0.05$, ****$p < 0.0001$) (from left to right $n = 413, 244, 243, 217, 256, 179,$ and $435$; $n$-values correspond to individual cells). Where indicated, an independent Kruskal–Wallis one-way test was performed comparing ORAI2-3-DKO cells to ORAI2-SKO and ORAI3-SKO cell lines. Scatter plots (**n–p**) are represented as mean ± SEM and were statistically analyzed using an ordinary one-way ANOVA with multiple comparisons to WT HEK293 (*$p < 0.05$).

We show that ORAI1/2/3 triple-knockout (ORAI-TKO) cells failed to sustain oscillations and displayed a similar behavior to that of WT HEK293 cells in which SOCE was blocked by inclusion of a low concentration of lanthanides (5 μM Gd³⁺) in the bath (Fig. 1b, c, f). This relatively low concentration of Gd³⁺ selectively blocks native SOCE and $I_{CRAC}$[23,24]. We show here that $I_{CRAC}$ mediated by each ORAI channel isoform independently co-expressed with STIM1 in ORAI-TKO cells is blocked by 5 μM Gd³⁺ (Supplementary Fig. 3). Furthermore, when we forced heteromeric ORAI channel formation by co-expressing STIM1 with ORAI concatenated dimers (ORAI1,1; ORAI1,2; ORAI1,3; and ORAI2,3; Supplementary Fig. 13b) in ORAI-TKO cells, $I_{CRAC}$ mediated by these different concatemers was also blocked by 5 μM Gd³⁺ (Supplementary Fig. 4), suggesting that all channels formed by ORAI homomeric or heteromeric combinations are blocked by 5 μM Gd³⁺. The incubation of ORAI-TKO cells with 5 μM Gd³⁺ had no additional effect on oscillations in these cells (Fig. 1d, f). The use of high concentrations of lanthanides (1 mM Gd³⁺), which block both Ca²⁺ entry and Ca²⁺ extrusion (so-called lanthanide insulation[25]) in ORAI-TKO cells restores their oscillatory behavior to levels of WT HEK293 cells (Fig. 1e, f), providing functional confirmation that ORAI-TKO cells have preserved IP₃R and SERCA functions. ORAI1 single-knockout HEK293 cells (ORAI1-SKO) showed almost complete abrogation of SOCE in response to thapsigargin (only an extremely small residual SOCE remains) as universally reported in essentially all

cell types (Fig. 1r, x). Surprisingly, however, ORAI1-SKO cells had preserved Ca²⁺ oscillations in response to 10 μM Cch, with similar oscillatory frequencies to those of WT HEK293 cells (Fig. 1g, m). Importantly, ORAI1-SKO cells responded exclusively with Ca²⁺ oscillations, showing no responses in the form of plateaus (Fig. 1o). ORAI1-SKO cells displayed an average of 86% of oscillating cells (Fig. 1n) with 14% of cells showing no response (Fig. 1p). One potential explanation for this paradoxical result is that ORAI2 and/or ORAI3 contribute discrete Ca²⁺ entry under low to mid agonist concentrations to support sustained oscillations in ORAI1-SKO cells.

**ORAI2 and ORAI3 are required for Ca²⁺ oscillations.** We then undertook a thorough side-by-side comparison between WT HEK293 cells and ORAI single isoform knockout cells, namely ORAI1-SKO, ORAI2-SKO, and ORAI3-SKO cells. Compared with WT HEK293 cells, ORAI2-SKO and ORAI3-SKO cells showed increased SOCE in response to 2 μM thapsigargin (Fig. 1s, t, x), as well as increased oscillatory frequency triggered by 10 μM Cch (Fig. 1h, i, m), although these increases were only statistically significant for ORAI2-SKO cells. Nevertheless, both ORAI2-SKO and ORAI3-SKO cells showed a significant increase in the % of cells with responses in the form of plateaus (Fig. 1o) and a decrease in the % of oscillating cells (Fig. 1n), with a more pronounced effect in ORAI3-SKO cells.

Next, we performed similar side-by-side comparisons between WT HEK293 cells and ORAI isoform double-knockout (DKO) cells, namely ORAI2,3-DKO, ORAI1,3-DKO, and ORAI1,2-DKO cells. SOCE in response to thapsigargin is abrogated in ORAI1,2-DKO and ORAI1,3-DKO cells (Fig. 1v, x), highlighting the crucial role for ORAI1 in mediating robust $Ca^{2+}$ responses, in the form of plateaus, to maximal store depletion. Conversely, SOCE in response to thapsigargin was increased in ORAI2,3-DKO cells compared with WT HEK293 cells (Fig. 1u, x) and this increase was significantly bigger than that of ORAI2-SKO and ORAI3-SKO cells (Fig. 1x), suggesting that the ORAI2- and ORAI3-negative regulations of SOCE are additive. We then performed direct ER $Ca^{2+}$ measurements using the ER-targeted genetically encoded indicator CEPIA1er in all three ORAI-DKO cells (Fig. 1q). Cells were treated by the reversible SERCA blocker cyclopiazonic acid (CPA; 50 μM) in 0 mM external $Ca^{2+}$ followed by washout of CPA and replenishment of 2 mM external $Ca^{2+}$ to measure ER store refilling. As expected, ORAI1,2-DKO and ORAI1,3-DKO have attenuated ER store refilling compared with WT HEK293 cells, whereas ORAI2,3-DKO showed faster and greater store refilling (Fig. 1q).

When stimulated with 10 μM Cch, ORAI1,2-DKO and ORAI1,3-DKO cells failed to show any responses in the form of plateaus (Fig. 1k, l, o) responding exclusively by producing regenerative $Ca^{2+}$ oscillations (Fig. 1n), with significantly enhanced proportion of non-responding cells in ORAI1,2-DKO cells (Fig. 1p). However, ORAI2,3-DKO cells showed a dramatic increase in % of cells exhibiting plateaus (Fig. 1o) with a decrease in % of oscillating cells (Fig. 1n). These changes in % of plateau and oscillating cells observed in ORAI2,3-DKO cells were more pronounced than those of ORAI2-SKO and ORAI3-SKO cells, again arguing that the effects on SOCE by ORAI2 and ORAI3 are additive.

We have until now recorded $Ca^{2+}$ oscillations only for the first 14 min after agonist addition. To document that $Ca^{2+}$ oscillations in cells lacking ORAI1 continue unabated for extended periods of time, we performed in a limited set of experiments, 1 h-long recordings of $Ca^{2+}$ oscillations in WT HEK293, ORAI1-SKO, ORAI1,3-DKO, and ORAI-TKO cells in response to 10 μM Cch (Supplementary Fig. 5). These data show that WT HEK293, ORAI1-SKO and ORAI1,3-DKO cells continue to oscillate for the duration of the recordings, whereas ORAI-TKO cells did not (Supplementary Fig. 5). Overall, these data show that cells expressing ORAI3 only (ORAI1,2-DKO) and ORAI2 only (ORAI1,3-DKO) readily show oscillatory behavior, whereas cells expressing ORAI1-only (ORAI2,3-DKO) predominantly plateau, suggesting that ORAI2 and ORAI3 are critical for $Ca^{2+}$ responses under modest, more physiological agonist concentrations, likely by acting as negative regulators of ORAI1, and that the native CRAC channel is likely a heteromultimer combining ORAI1 with ORAI2 and/or ORAI3.

To this point, we analyzed ORAI-SKO and DKO cells with one agonist concentration (10 μM Cch). To gain better insights into ORAI isoform contribution in response to a wide range of agonist concentrations, we measured $Ca^{2+}$ signals in response to increasing concentrations of agonist ranging from 1 to 30 μM (Fig. 2a–n), focusing on the % of cells that show oscillations vs. the % of cells showing plateaus. ORAI2-SKO and ORAI3-SKO cells were more prone to manifest oscillations at the lowest agonist doses (1–3 μM), whereas showing the least % of oscillating cells at the highest dose (30 μM; Fig. 2c, d, i), with ORAI3-SKO cells showing higher % of oscillating cells than those of ORAI2-SKO cells. However, ORAI1-SKO generated almost no oscillations at low doses (1–3 μM) and a robust % of oscillating cells at the highest doses (10–30 μM; Fig. 2b, i). As for plateau responses, ORAI1-SKO cells failed to support plateaus at all

concentrations, whereas ORAI2-SKO and ORAI3-SKO cells showed significantly enhanced plateaus responses compared with WT HEK293 cells, with ORAI2-SKO cells showing higher % of plateaus responses at all concentrations (Fig. 2j). For comparisons, the % of non-responding cells are shown in Fig. 2k.

Next, we used the same protocol to analyze ORAI-DKO cells. ORAI1,2-DKO and ORAI1,3-DKO cells showed a similar % of oscillating cells to WT cells at all concentrations of agonist (Fig. 2l). However, ORAI2,3-DKO-cells showed a higher % of oscillating cells compared with WT cells at 1–3 μM Cch, but this % significantly decreased at higher concentrations (10–30 μM; Fig. 2l). Conversely, ORAI1,2-DKO and ORAI1,3-DKO cells failed to mediate plateaus responses at all concentrations while ORAI2,3-DKO cells showed enhanced % of plateaus cells compared with WT at all concentrations (Fig. 2m). For comparisons, the % of non-responding cells are shown in Fig. 2n. Representative traces of $Ca^{2+}$ oscillations from five different cells/condition for the double ORAI-knockout cells (ORAI2,3-DKO, ORAI1,3-DKO, and ORAI1,2-DKO) and the ORAI-TKO cells are shown in Fig. 2e–h. ORAI-TKO cells responded mostly by one broad $Ca^{2+}$ spike in response to each increasing concentration of agonist (Fig. 2h). These data suggest that although all three ORAI isoforms can support $Ca^{2+}$ oscillations on their own, ORAI2 and ORAI3 dampen the strength of the $Ca^{2+}$ signal at all concentrations of agonists. These experiments strongly argue that the native CRAC channel is constructed through heteromultimerization of ORAI1 with ORAI2 and ORAI3.

An earlier report used small interfering RNA (siRNA) knockdown of ORAI1 in HEK293 cells and proposed that ORAI1 is required for mediating $Ca^{2+}$ oscillations[25]. We considered that the discrepancy between this report and our own findings could be due to differences between complete ORAI1-knockout in our CRISPR/Cas9 clones versus partial knockdown with siRNA. We therefore tested the effect of siRNA-mediated ORAI1 knockdown on $Ca^{2+}$ oscillations in our parental HEK293 cells. We documented ORAI1 knockdown at the protein level on transfected cell populations (Fig. 2o, p). We then employed a comprehensive protocol where $Ca^{2+}$ oscillations for each individual cell can be directly compared with the corresponding maximal SOCE activity of the same cell. $Ca^{2+}$ oscillations were triggered by addition of 10 μM Cch and recorded for the first 15 min followed by addition of thapsigargin to determine maximal SOCE activity (Fig. 2q). Our data clearly show that, in HEK293 cell populations transfected with siRNA against ORAI1, SOCE activity in response to thapsigargin was inhibited by ~65% (Fig. 2r), matching closely the extent of ORAI1 protein knockdown (Fig. 2o, p). However, in the same cells in which ORAI1 was knocked-down by siRNA and SOCE was decreased, we recorded normal frequencies of $Ca^{2+}$ oscillations indistinguishable from HEK293 cells transfected with non-targeting siRNA (Fig. 2q, s). ORAI1 knockdown caused a slight increase in the % of oscillating cells (Fig. 2t), an increase in the % of non-responding cells (Fig. 2v), and reduced % of plateau cells (Fig. 2u). These results are consistent with data obtained with ORAI1-SKO clones generated by CRISPR/Cas9 technology.

To provide further evidence for lack of off-target effects in different KO clones, we used our ORAI-TKO cells to perform independent rescue experiments of each individual ORAI isoform. Previous studies showed that under conditions of ectopic ORAI isoform co-expression with STIM1 in WT HEK293 cells (both driven by the strong cytomegalovirus (CMV) promoter), the magnitude of SOCE activated by thapsigargin is as follows: ORAI1 > ORAI2 > ORAI3[11,12]. To obtain physiologically meaningful results that relate to native conditions of expression, we expressed each individual ORAI isoform in ORAI-TKO cells using plasmids that achieve expression levels similar to the native condition. These plasmids were generated by sub-cloning the

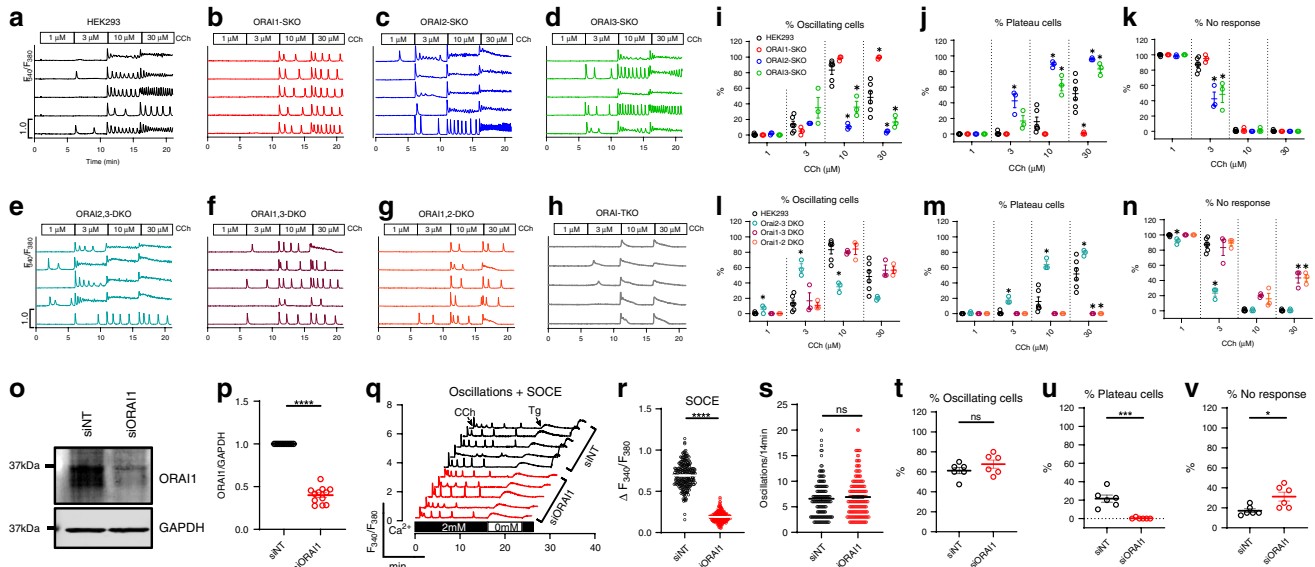

**Fig. 2 ORAI3 and ORAI2 are required for Ca²⁺ oscillations. a–h** Representative Ca²⁺ oscillations in response to increasing concentrations of carbachol (1–30 μM Cch) measured using Fura2 in parental HEK293 cells, and in single, double, and triple ORAI knockout cells. Cells were maintained in HBSS containing 2 mM Ca²⁺ for the duration of the experiments and stimulated with 1, 3, 10, and 30 μM Cch for a duration of 5 min each. Representative traces were chosen from five cells/condition to represent the datasets as a whole. **i–n** Quantification of % of oscillating cells (**i**, **l**), % of plateau cells (**j**, **m**), and % of non-responding cells (**k**, **n**) for conditions from **a–h** (for **i–k** and **l–n**, n = 6, 3, 3, and 3 from left to right; each point represents an independent experiment). Quantification of oscillation data was performed using one-way ANOVA with multiple comparisons to WT HEK293 cells (*p < 0.05). **o** Representative Western blot documenting ORAI1 protein knockdown with siRNA in WT HEK293 cells. **p** Quantification of ORAI1 protein knockdown using ORAI1/GAPDH densitometry. Each data point represents a technical replicate from four independent knockdown experiments (n = 12). **q** Representative Ca²⁺ oscillations in response to 10 μM carbachol (Cch) followed by measurements of SOCE elicited by subsequent addition of 2 μM thapsigargin in HEK293 cells transfected with siRNA against ORAI1 (siORAI1) and non-targeting control siRNA (siNT). **r** Quantification of the magnitude of SOCE in siNT- and siORAI1-transfected cells (n = 240 for siNT and 256 for siORAI1; each point represents an individual cell). **s–v** Quantification of total oscillations/14 min (**s**), % of oscillating cells(n = 144 for siNT and 182 for siORAI1; each point represents an individual cell) (**t**), % of plateau cells (**u**), and % of non-responding cells (**v**) in siNT- and siORAI1-transfected cells (for **t–v**, n = 6 for both siNT and siORAI1; each point represents an independent experiment). All siRNA scatter plots, which represent the mean ± SEM were analyzed using the Mann–Whitney U-test (*p < 0.05; ***p < 0.001; ****p < 0.0001; ns, not significant).

yellow fluorescent protein (YFP)-tagged cDNA of each ORAI isoform under the control of the "weak" thymidine kinase (tk) promoter[26]. Hence, we circumvented the need to overexpress STIM1 and relied on native STIM proteins to activate these ORAI isoforms (see also ref. [21]). Based on YFP fluorescence of tagged ORAI isoforms, we analyzed only cells with comparable levels of YFP fluorescence (Fig. 3d, k). We would like to stress however that although we ensured that all ORAI isoforms were rescued to similar levels in ORAI-TKO cells, native ORAI2 and ORAI3 levels are likely significantly lower than those of ORAI1 in WT HEK293 cells. Indeed, the use of thapsigargin to maximally activate SOCE in ORAI-TKO cells expressing individual low-expressing tk-driven ORAIs showed that ORAI2 and especially ORAI3 mediate significant SOCE compared with ORAI1,3-DKO and ORAI1,2-DKO cells, respectively (i.e., cells expressing native ORAI2 or native ORAI3 only; compare Fig. 3a–c with Fig. 1v–x).

In response to 10 μM Cch, each ORAI isoform was capable of supporting oscillations in individual cells with frequencies of oscillations marginally higher in ORAI2- and ORAI3-expressing ORAI-TKO cells than ORAI1-expressing cells (Fig. 3e–h). Significantly, ORAI2- and ORAI3-expressing cells showed the highest % of oscillating cells (Fig. 3i), whereas ORAI1-expressing cells showed the highest % of cells with plateaus (Fig. 3j). Although the overall trend of these experiments matched results obtained with ORAI DKO cells (i.e., cells expressing only one native ORAI; Fig. 1j–p), there were significant differences in the proportions of oscillating cells vs. plateau cells, suggesting again

that native expression of ORAI2 and ORAI3 in HEK293 cells is significantly lower than that of ORAI1.

**ORAI2 and ORAI3 interact with STIM1 under basal conditions.** To provide insight into the requirement of ORAI2 and ORAI3 for sustaining Ca²⁺ oscillations, we tested whether STIM1 preferentially interacts with ORAI2 and ORAI3 over ORAI1. We reasoned that a potential privileged interaction of ORAI2 and ORAI3 with STIM1 would allow these two ORAI isoforms to be activated by lower levels of store depletion, such as those evoked by low agonist concentrations. Interestingly, Förster resonance energy transfer (FRET) measurements of STIM1-YFP co-expressed with either CFP-ORAI isoform in ORAI-TKO cells showed that ORAI2 and ORAI3 have a significantly higher basal FRET with STIM1 compared with ORAI1 (Fig. 3l, m). Store depletion with thapsigargin caused enhanced FRET between STIM1 and all three ORAI isoforms. Surprisingly however, the maximal FRET level induced by thapsigargin was larger and faster for ORAI2/STIM1 compared with ORAI3/STIM1, with ORAI1/STIM1 showing intermediary values (Fig. 3l, n, o). Figure 3p documents that FRET imaging was performed in cells with values of YFP/cyan fluorescent protein (CFP) fluorescence ratios of ~1.

**The native CRAC channel is an ORAI heteromultimer.** Previous studies have demonstrated that ORAI isoforms co-expressed

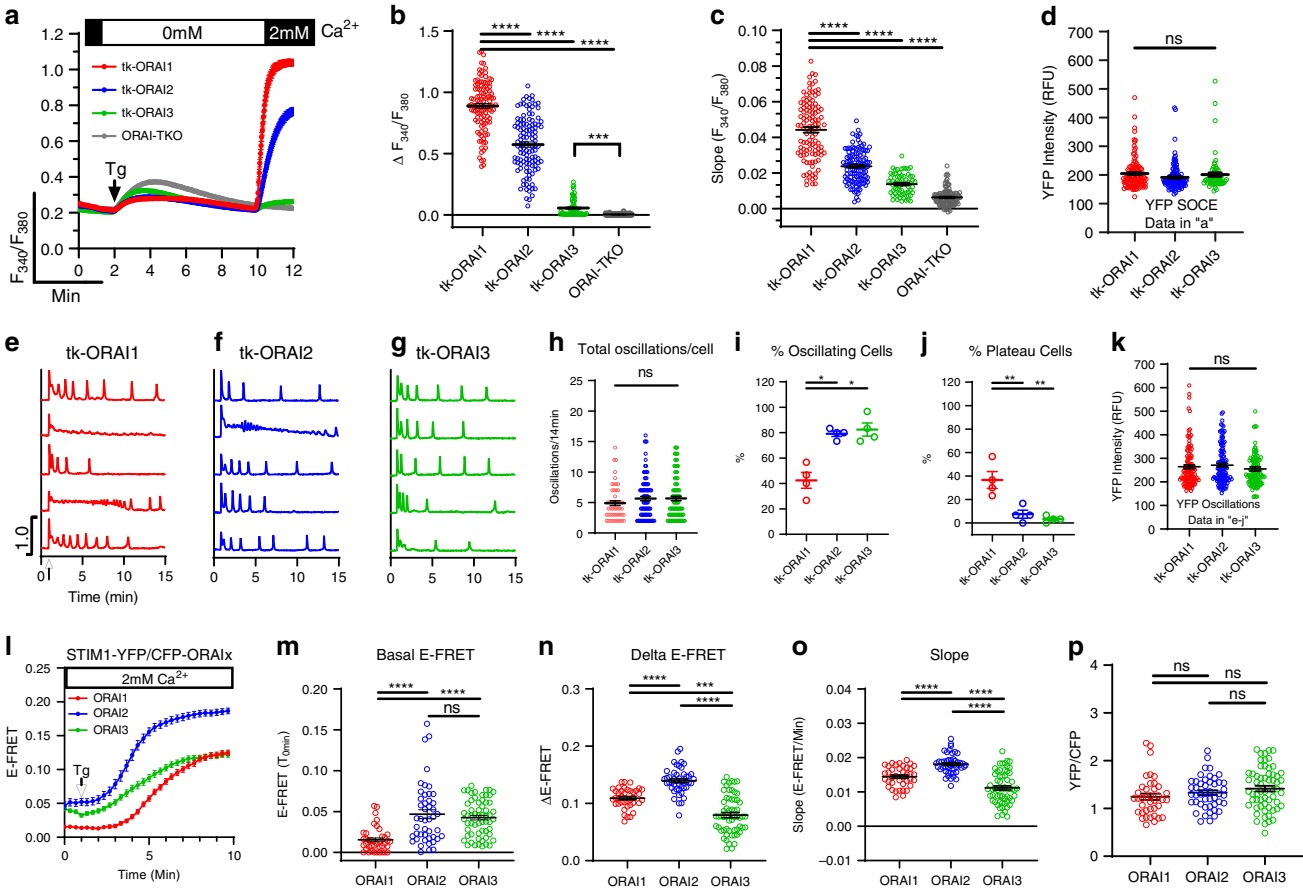

**Fig. 3 ORAI3 and ORAI2 interact with STIM1 under basal conditions. a** Measurements of ER $Ca^{2+}$ release (in 0 mM $Ca^{2+}$) and SOCE (in 2 mM $Ca^{2+}$) upon store depletion with 2 μM thapsigargin (Tg) in ORAI-TKO cells (untransfected controls) and in ORAI-TKO cells individually rescued with either ORAI1, ORAI2, or ORAI3 driven by the "weak" thymidine kinase (tk) promoter; all traces are plotted as mean ± SEM. **b–d** Scatter blot representing quantification of the magnitude (**b**) and slope (**c**) of SOCE and fluorescence intensity (**d**) of transfected tk-driven YFP-ORAI isoforms in **a** (for **b–d** from left to right $n = 112, 116, 67$, and 120; n-values correspond to individual cells). **e–g** Representative $Ca^{2+}$ oscillations in response to 10 μM carbachol (Cch; added where indicated by arrow) measured using Fura2 over the course of 15 min in ORAI-TKO cells transfected with individual tk-driven ORAI isoforms. Cells were maintained in HBSS containing 2 mM $Ca^{2+}$ for the duration of the experiments and stimulated with 10 μM Cch at 1 min (as indicated by arrow in "**e**"). Representative traces from five cells/condition were chosen to represent the datasets as a whole. **h–j** Quantification of total oscillations/14 min (**h**), % of oscillating cells ($n = 51$(ORAI1), 94(ORAI2), and 100(ORAI3); data represent individual cells) (**i**), and % of plateau cells (**j**) for data in **e–g** (for **i**, **j**, $n = 4$ for all conditions; data represent independent experiments). **k** Fluorescence intensity of transfected tk-driven YFP-ORAI isoforms for data in **e–j** ($n = 120$ (ORAI1), 119 (ORAI2), and 118 (ORAI3); data represent individual cells). **l** E-FRET represented as mean ± SEM between STIM1-YFP and either CFP-ORAI isoform at rest and after addition of 2 μM thapsigargin (Tg) in HBSS containing 2 mM $Ca^{2+}$. **m–p** Scatter blots representing quantification of basal E-FRET (**m**), the magnitude (**n**), and slope (**o**) of E-FRET change after addition of thapsigargin and YFP/CFP fluorescence ratios (**p**) from data in **l** (for **m–p**, $n = 39$ (ORAI1), 46 (ORAI2), and 60 (ORAI3); data represent individual cells). All data are represented as mean ± SEM. Panels **b–d**, **h**, **k**, **m–p** were statistically analyzed using the Kruskal–Wallis one-way ANOVA with multiple comparisons. Panels **i**, **j** were analyzed using one-way ANOVA with multiple comparisons (*$p < 0.05$; **$p < 0.01$; ***$p < 0.001$; ****$p < 0.0001$; ns, not significant). All comparisons were made to tk-ORAI1.

with STIM1 in WT HEK293 cells generate CRAC currents that significantly differ in their fast CDI, a mechanism of $Ca^{2+}$-mediated negative feedback that limits $Ca^{2+}$ entry[10]. The strength of CDI is the highest for ORAI3, lowest for ORAI1 and intermediate for ORAI2 (ORAI3 > ORAI2 > ORAI1)[11,12]. These differences in fast CDI between ORAI isoforms were confirmed in our co-expression studies, now performed in ORAI-TKO cells and showing that CDI of ORAI3 > ORAI2 > ORAI1, whereas peak whole-cell currents follow an inverse relationship of ORAI1 > ORAI2 > ORAI3 (Fig. 4a–e). Compared with ORAI1, the higher basal interactions of ORAI2 and ORAI3 with STIM1 (Fig. 3m–q) and the higher CDI of these two isoforms, provide a plausible mechanism of how native ORAI2 and ORAI3 heteromerization with ORAI1 generates CRAC channels that are both activated by modest levels of store depletion and possess reduced activity.

In support of ORAI isoform heteromerization, we show that co-expression of STIM1 with combinations of two ORAI isoforms (Fig. 4f–j) or ORAI concatenated heterodimer constructs (Fig. 4k–o) generates CRAC currents with an intermediary CDI and peak current densities that represent hybrids of those of the individual ORAI isoforms. Peak currents generated by expression of ORAI concatenated dimer constructs are smaller, likely reflecting a relatively lower expression of these constructs. We provide FRET evidence showing that co-expression of two ORAI isoforms (ORAI1/ORAI2, ORAI1/ORAI3, or ORAI2/ORAI3) in ORAI-TKO cells lead to robust basal FRET interactions between these ORAI isoforms, suggesting ORAI isoforms readily interact. In these FRET experiments, STIM1-mCherry was co-transfected in each experimental condition to ensure optimal activation of overexpressed ORAI channel isoforms. These strong basal FRET between ORAI isoform pairs

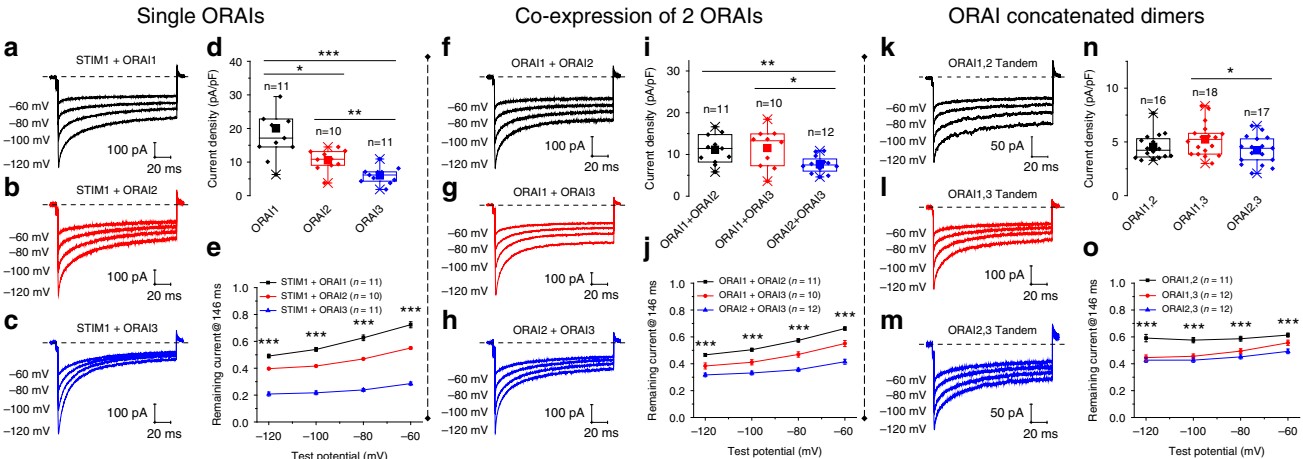

**Fig. 4 ORAI isoforms form heteromeric channels. a–c** Representative currents were recorded with a pipette solution containing 10 mM EGTA and elicited by hyperpolarizing voltage steps (holding potential +30 mV) from ORAI-TKO cells co-expressing YFP-STIM1 (4 µg plasmid) with individual CFP-ORAI isoforms (1 µg plasmid). **d, e** Peak current densities at −100 mV are shown in **d** and the extent of ORAI isoform CDI represented as current remaining at 146 ms are shown in **e**; each data point represents mean ± SEM. **f, g** Representative currents were recorded with a pipette solution containing 10 mM EGTA and elicited by hyperpolarizing voltage steps (holding potential +30 mV) from ORAI-TKO cells co-expressing YFP-STIM1 (4 µg plasmid) with two CFP-ORAI isoforms (0.5 µg plasmid each). **i, j** Peak current densities at −100 mV are shown in **i** and the extent of CDI of ORAI isoform combinations are represented as current remaining at 146 ms and shown in **j**; each data point represents mean ± SEM. **k–m** Representative currents were recorded with a pipette solution containing 10 mM EGTA and elicited by hyperpolarizing voltage steps (holding potential +30 mV) from ORAI-TKO cells co-expressing YFP-STIM1 (4 µg plasmid) with 1 µg plasmid of concatenated ORAI isoform heterodimers C-terminally tagged with tdTomato (e.g., ORAI1,2 Tandem). **n, o** Peak current densities at −100 mV are shown in **n** and the extent of CDI of ORAI concatemers are represented as current remaining at 146 ms and shown in **o**; each data point represents mean ± SEM. All data were statistically analyzed using one-way ANOVA with multiple comparisons (*$p < 0.05$; **$p < 0.01$; ***$p < 0.001$; ****$p < 0.0001$). Comparisons deemed significant are indicated by horizontal bars. Boxplots show the mean, median, and the 75th to 25th percentiles.

were not further enhanced by agonist (10–100 µM Cch) or by maximal store depletion with 1 µM ionomycin (Supplementary Fig. 6).

To provide evidence that ORAI isoform heteromerization occurs under native conditions, we transfected either WT HEK293 cells or ORAI-SKO cells with 3 µg plasmid cDNA of various ORAI pore mutants and measured both SOCE (in response to maximal store depletion with thapsigargin; Fig. 5a–e) and $Ca^{2+}$ oscillations triggered by 10 µM Cch (Fig. 5f; see also Supplementary Fig. 7). These ORAI pore mutants are: the ORAI1 pore mutants (E106Q or E106A), the ORAI2 pore mutant (E80Q) and the ORAI3 pore mutant (E81Q). The expression of either of these pore mutants in WT HEK293 cells causes dominant negative effects, essentially abrogating native SOCE and $I_{CRAC}$[27–29]. As expected, we show that all pore mutants essentially abrogated SOCE in WT HEK293 cells as well as the small residual SOCE in ORAI1-SKO cells (Fig. 5a–e). Surprisingly, $Ca^{2+}$ oscillations in cells expressing pore mutant ORAIs were only partially inhibited, with ORAI1-E106A, ORAI2-E80Q, and ORAI3-E81Q producing greater inhibitory effect than ORAI1-E106Q on both the frequency of $Ca^{2+}$ oscillations (Fig. 5f) and on the % of oscillating cells (Supplementary Fig. 7). Recent studies showed that co-expression of STIM1 with a concatenated heterodimer consisting of WT ORAI1 and ORAI1-E106A in ORAI-SKO cells generated SOCE and membrane currents that were substantially reduced compared with a concatenated heterodimer composed of WT ORAI1 and ORAI1-E106Q[30]. These results suggest that ORAI isoforms heteromerize under native conditions and that $Ca^{2+}$ oscillations can be sustained by a minuscule amount of $Ca^{2+}$ entry that is below the detectable limit of Fura2.

Further evidence for native ORAI isoform heteromultimerization was gleaned from comparative measurements of native CRAC currents in WT HEK293 cells in our single and double ORAI-knockout cell lines. Due to the very small size of native CRAC

currents, we used divalent-free (DVF) bath solutions to amplify these currents in WT HEK293 cells and ORAI-KO clones (Fig. 5g–m; upper panel); the $I/V$ curves (black traces) are shown in the middle panels while data summary of current densities from several recordings are shown in the lower panels. We exploited the different pharmacological sensitivity of ORAI isoforms to the compound 2-aminoethoxydiphenyl borate (2-APB), whereby at 50 µM 2-APB, ORAI1 is completely inhibited, ORAI2 is only partially inhibited, and ORAI3 is potentiated[31] (see also Supplementary Fig. 8 for CRAC recordings from HEK-TKO cells co-expressing STIM1 with each ORAI isoform). Native CRAC currents are readily measured in WT HEK293 cells and are blocked by 50 µM 2-APB or 10 µM GSK-7975A (Fig. 5g and Supplementary Fig. 9; red $I/V$ curve in middle panel). However, in ORAI-SKO and ORAI-DKO cells, CRAC currents are only observed in cells expressing ORAI1 (Fig. 5g, i–k), consistent with ORAI1 having the highest overall activity and the highest expression level in native cells. In single and double ORAI-KO clones, 2-APB-mediated CRAC current potentiation is detected only in cells that have preserved ORAI3 expression but lack expression of either ORAI1 (Fig. 5h) or ORAI2 (Fig. 5i), or both (Fig. 5m), arguing that ORAI isoforms form heteromultimers under native conditions. Please note that in the case of ORAI2-SKO cells (Fig. 5i), native CRAC currents are unaffected by addition of 50 µM 2-APB, presumably because of concomitant inhibition of native ORAI1 and activation of native ORAI3.

**Mathematical model supports CRAC as an ORAI heteromultimer.** We constructed a simplified model of how ORAI1, ORAI2, and ORAI3 affect $Ca^{2+}$ signals via their control of $Ca^{2+}$ entry across the PM. The model of $Ca^{2+}$ oscillations is based on a model published previously[22] and assumes a well-mixed cell with deterministic IP3Rs. In the model, $Ca^{2+}$ influx is a decreasing

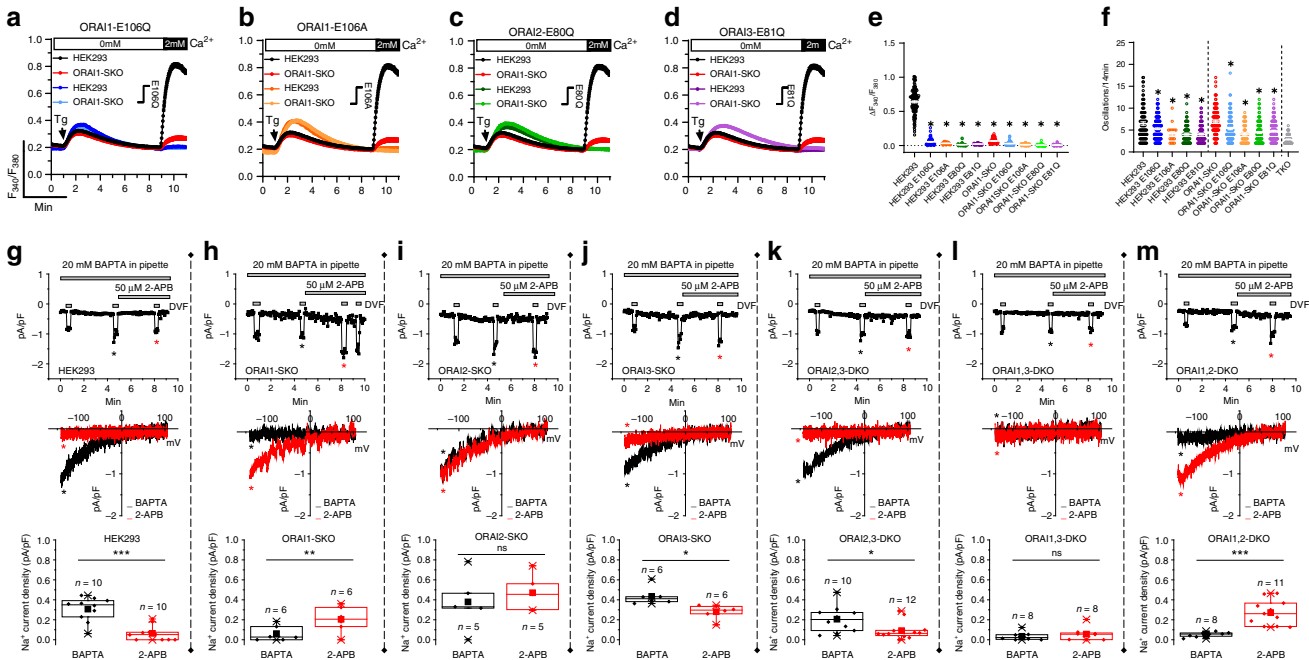

**Fig. 5 Native CRAC channels are heteromultimers of ORAI isoforms. a–d** Measurements of ER Ca²⁺ release (in 0 mM Ca²⁺) and SOCE (in 2 mM Ca²⁺) on store depletion with 2 μM thapsigargin (Tg) in HEK293 cells and ORAI1 single-knockout (ORAI1-SKO) cells either untransfected or transfected with ORAI pore mutants: ORAI1-E106Q (**a**), ORAI1-E106A (**b**), ORAI2-E80Q (**c**), and ORAI3-E81Q (**d**). **e** Quantification of SOCE magnitude for **a–d** (left to right $n = 130, 76, 68, 64, 53, 116, 53, 57, 51$, and $37$; points represent individual cells). **f** Quantification of oscillations/14 min for **a–d**; data from ORAI-TKO cells (Fig. 1f) is included as a reference (left to right $n = 97, 142, 175, 124, 119, 119, 127, 145, 19, 70$, and $84$; points represent individual cells). Data are represented as mean ± SEM and were statistically analyzed using the Kruskal–Wallis one-way ANOVA with multiple comparisons (*$p < 0.05$). Comparisons in **e** were made to WT HEK293. Comparisons in **f** were made to the corresponding untransfected control (i.e., WT HEK293 or ORAI1-SKO). **g–m** CRAC current measurements on store depletion by dialysis of 20 mM 1,2-bis(o-aminophenoxy)ethane-N,N,N′,N′-tetraacetic acid (BAPTA) through the pipette followed by perfusion of 50 μM 2-APB in the bath. Native CRAC current recordings were performed in HEK293 cells (**g**), ORAI1-SKO cells (**h**), ORAI2-SKO cells (**i**), ORAI3-SKO cells (**j**), ORAI2,3-DKO cells (**k**), ORAI1,3-DKO cells (**l**), and ORAI1,2-DKO cells (**m**). Top panels: representative current traces elicited by voltage ramps from −140 mV to +100 mV from a holding potential of +30 mV with three pulses of divalent-free solutions to amplify native CRAC currents. The first pulse is performed immediately after whole-cell configuration, the second pulse after CRAC current has been maximally activated (4–5 min after break-in) and the third pulse after 2-APB perfusion (~8 min after break-in). (middle panels) representative $I/V$ curves for maximal CRAC current (black trace) and after 2-APB addition (red trace) taken where indicated in top panels by the color-coded asterisks. (lower panels) statistical analysis of peak background-subtracted Na⁺ CRAC current densities (represented as mean ± SEM) recorded from several cells in DVF solutions before and after 2-APB perfusion. Data were statistically analyzed using a two-tailed Student's $t$-test (*$p < 0.05$; **$p < 0.01$; ***$p < 0.001$; ns, not significant). Boxplots show the mean, median, and the 75th to 25th percentiles.

sigmoidal function of ER [Ca²⁺]. To simplify the combinatorial aspects, we assumed that only ORAI dimers form; the possible effects of, say, ORAI3 inhibition of ORAI1 are incorporated via the parameters that set the conductance of the various ORAI dimers. Full model details and some typical model simulations are given in the methods section, the results of these simulations can be seen in Fig. 6 and Supplementary Fig. 10. Although there are not enough data to fit all the model parameters unambiguously, we found that the model could reproduce the experimental data (Figs. 1 and 2) only when, first, ORAI2 and ORAI3 dimerize preferentially with ORAI1, rather than with themselves; second, that the ORAI1,2 and ORAI1,3 heterodimers allow substantially less Ca²⁺ current than does the ORAI1,1 homodimer; and lastly, that there is two- to fivefold more ORAI1 than there is ORAI2 or ORAI3. All these predictions fit our experimental data described above. Figure 6a–c shows model predictions where the parameter $p$ (with units of μM) is equivalent to agonist concentration over time (s). Figure 6d shows model predictions of responses to thapsigargin in the absence then presence of external Ca²⁺. The response to thapsigargin is simulated by decreasing the SERCA pump flux to zero. The differences between ORAI2 and ORAI3 in their negative regulation of ORAI1 and Ca²⁺ oscillations cannot be distinguished by the model, and as such the model predictions

depicted in Fig. 6 have omitted ORAI3-SKO and ORAI1,3-DKO cells, which are assumed to be the same as ORAI2-SKO and ORAI1,2-DKO cells, respectively (Fig. 6b, c). The Ca²⁺ responses predicted in response to increasing concentrations of agonist generate plateaus earlier in ORA2,3-DKO cells (Fig. 6c) compared with WT HEK293 cells (Fig. 6a). The model also predicts that maximal store depletion by thapsigargin produced almost no SOCE response in ORAI1,2-DKO cells, whereas SOCE is enhanced in ORAI2-SKO cells and further enhanced in ORAI2,3-DKO cells (Fig. 6d). In particular, reintroduction of external Ca²⁺ in the ORAI2,3-DKO case gives a response that is considerably larger than WT, as ORAI1 is now relieved from inhibition by ORAI2 and ORAI3. For all above cell lines, we have also modeled ER Ca²⁺ content in response to increasing concentrations of agonist and to thapsigargin (Tg) stimulation as previously described (Supplementary Fig. 10). These mathematical predictions are in agreement with our experimental findings, although there are some clear quantitative discrepancies, particularly in the initial response to thapsigargin.

**ORAI2/ORAI3 ensure differential activation of NFAT isoforms.** The translocation of NFAT isoforms to the nucleus

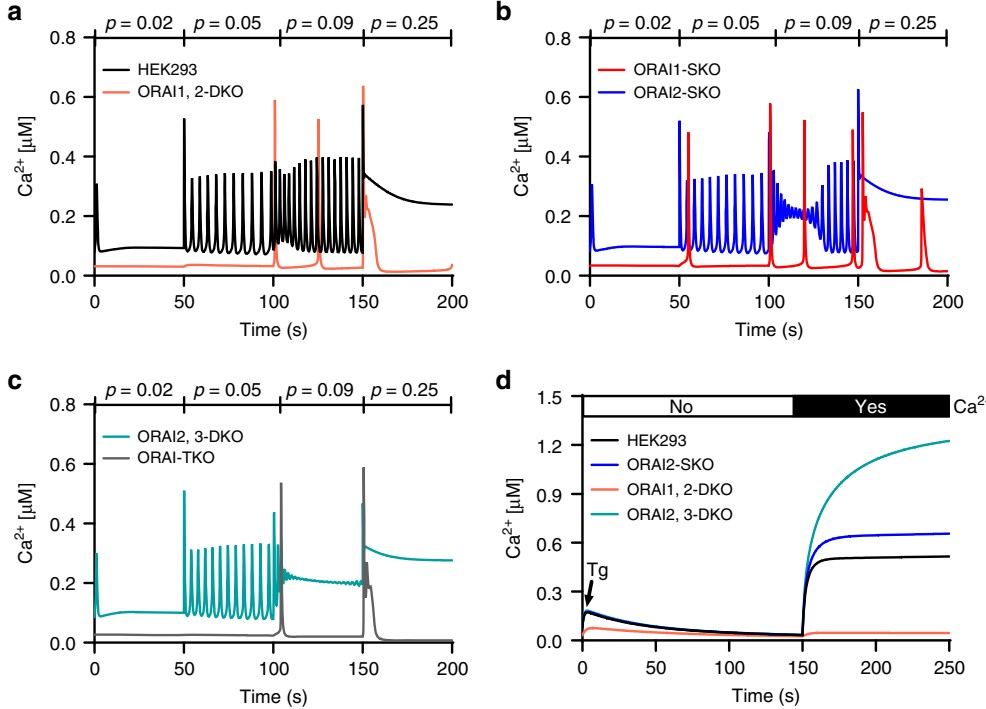

**Fig. 6 Mathematical modeling shows native CRAC channels are ORAI heteromultimers.** Model simulations used the parameter $p$ (with units of μM) as the equivalent to increasing agonist concentrations. Other parameter values are $I_1 = 1$, $I_2 = 0.2$, $I_3 = 0.2$ (all in units of concentration), $K_{11} = 14$, $K_{22} = 1.5$, $K_{33} = 0.93$, $K_{12} = 19$, $K_{13} = 7$, $K_{23} = 14$ (all in units of 1/concentration), and $\alpha_{11} = 2$, $\alpha_{22} = 1$, $\alpha_{33} = 0.7$, $\alpha_{12} = 0.1$, $\alpha_{13} = 0.1$, $\alpha_{23} = 0.1$ (all in units of 1/time). All other parameter values are as in ref. [22]. **a–c** Model the response to increasing agonist concentrations in parental HEK293 cells and 5 different single, double, and triple ORAI-knockout cell lines. **d** Models the cytosolic $Ca^{2+}$ response to maximal store depletion with thapsigargin in the absence then presence of 2 mM external $Ca^{2+}$.

depends on $Ca^{2+}$ entry specifically through SOCE. NFAT1 nuclear translocation occurs slowly and requires robust $Ca^{2+}$ entry through SOCE, whereas NFAT4 nuclear translocation occurs with more modest levels of SOCE activity such as when cells are stimulated with low concentrations of agonist[21,32–34]. We expressed green fluorescent protein (GFP)-tagged NFAT reporter constructs (NFAT1-GFP and NFAT4-GFP) in our parental WT HEK293 cell line and in each of our single, double and triple ORAI clones and then stimulated these cells with 10 μM Cch. Nuclear translocation of each NFAT-GFP fusion protein was recorded over time by monitoring the ratio of nuclear to whole-cell GFP fluorescence (Fig. 7). We show that both NFAT1 (Fig. 7a–g) and NFAT4 (Fig. 7h–n) translocation to the nucleus was significantly faster and higher in ORAI2-SKO, ORAI3-SKO, and ORAI2,3-SKO cells when compared with WT HEK293 cells, essentially causing significant NFAT1 activation at lower agonist concentrations that normally cause selective activation of the faster isoform NFAT4. Interestingly, NFAT1 and NFAT4 nuclear translocation was abrogated in all cells lacking ORAI1 (ORAI1-SKO, ORAI1,2-DKO, and ORAI1,3-DKO cells), including ORAI-TKO cells (Fig. 7d–g, k–n). These data suggest that (1) NFAT isoform nuclear translocation can only be supported by ORAI1 activity, and (2) ORAI2 and ORAI3 broaden the range of agonist sensitivity to allow differential activation of NFAT1 and NFAT4 at their respective ranges of agonist concentrations. A significant implication from these findings is as follows: despite preserved agonist-mediated $Ca^{2+}$ oscillations in ORAI1-SKO cells that are of similar frequency to WT HEK293 cells (see Figs. 1 and 2), these $Ca^{2+}$ oscillations are incapable of causing NFAT1 (Fig. 7a–c) and NFAT4 (Fig. 7h–j) nuclear translocation. Our results strongly suggest that specific $Ca^{2+}$ entry through ORAI1 is the signal required for NFAT isoform translocation and that, through

heteromerization with ORAI1, ORAI2, and ORAI3 broaden the bandwidth of agonist sensitivity to ensure selective nuclear translocation of each NFAT isoform in response to its physiological range of agonist concentrations.

## Discussion
In this study, we used single, double, and triple gene deletion of native ORAI channels and determined that by working together, the ORAI channel trio shapes the cytosolic $Ca^{2+}$ signal in response to wide ranges of agonist concentrations. Data from our rescue experiments of single tk-driven ORAIs in ORAI-TKO cells and data from single and double KO cells suggest that each ORAI isoform can support a specific range of $Ca^{2+}$ signaling events: ORAI3 and ORAI2 are critical for low- and mid-range oscillatory responses, whereas ORAI1 mediates high-range plateaus. We show that ORAI2 and ORAI3 interact with STIM1 under basal conditions. This observation coupled with the fact that ORAI2 and ORAI3 have greater CDI compared with ORAI1 suggest that ORAI2 and ORAI3 are tailored to mediate responses to the low and mid-range of agonist concentrations. Interestingly, despite the clear role for ORAI2 and ORAI3 in $Ca^{2+}$ oscillations in response to modest agonist concentrations, maximal store depletion with thapsigargin suggests that native ORAI2 and ORAI3 mediate only a miniscule amount of $Ca^{2+}$ entry, i.e., below the detection level of current methods. These data and the finding that ORAI pore mutants block SOCE in response to thapsigargin but only partially inhibit $Ca^{2+}$ oscillations argue that only a very small amount of SOCE activity is required to sustain $Ca^{2+}$ oscillations.

Our ORAI rescue experiments in ORAI-TKO cells suggest that ORAI2 and ORAI3 levels in HEK293 cells are significantly less

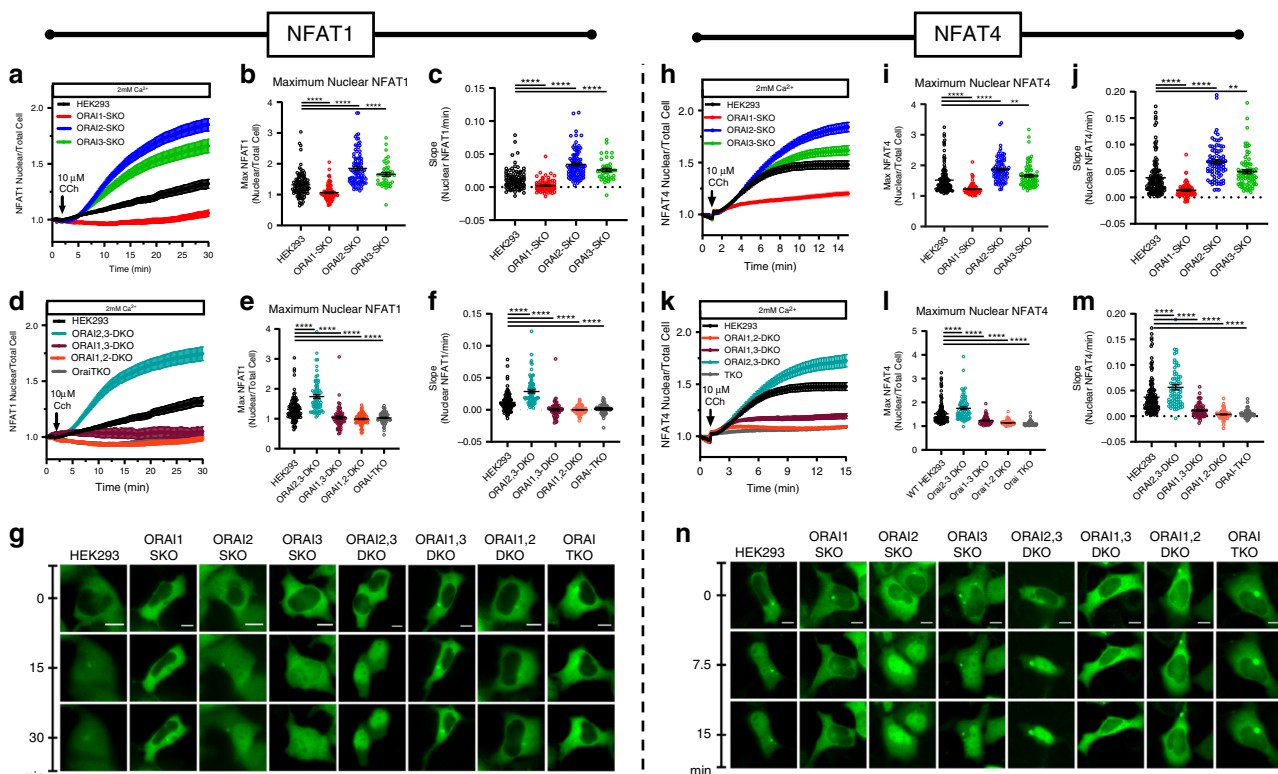

**Fig. 7 ORAI2 and ORAI3 ensure differential activation of NFAT isoforms. a**, **d** Time-lapse quantification represented as mean ± SEM of nuclear translocation in response to 10 μM Cch stimulation in the presence of 2 mM Ca$^{2+}$ of NFAT1-GFP transfected either in parental HEK293 cells or ORAI single- (**a**), double- (**d**), or triple- (**d**) knockout cells; images were captured every 10 s. **b**, **e** Maximal NFAT1-GFP nuclear translocation determined by the ratio of nuclear/total cell GFP fluorescence. **c**, **f** Rate of NFAT1-GFP nuclear translocation in response to Cch stimulation. **g** Representative images of different cell types transfected with NFAT1-GFP at time 0 and 15 min and 30 min after stimulation with 10 μM Cch. Scale bar: 10 μm (for all NFAT1 experiments n = 98(HEK293), n = 84(ORAI1-SKO), n = 93(ORAI2-SKO), n = 44(ORAI3-SKO), n = 79(ORAI1,2-DKO), n = 65(ORAI1,3-DKO), n = 76 (ORAI2,3-DKO), n = 64(ORAI-TKO); all points are single cells). **h**, **k** Time-lapse quantification represented as mean ± SEM of nuclear translocation in response to 10 μM Cch stimulation in the presence of 2 mM Ca$^{2+}$ of NFAT4-GFP transfected either in parental HEK293 cells or ORAI single- (**h**), double- (**k**), or triple- (**k**) knockout cells; images were captured every 10 s. **i**, **l** Maximal NFAT4-GFP nuclear translocation determined by the ratio of nuclear/total cell GFP fluorescence. **j**, **m** Rate of NFAT4-GFP nuclear translocation in response to Cch stimulation. **n** Representative images of different cell types transfected with NFAT4-GFP at time 0, and 15 min and 30 min after stimulation with 10 μM Cch. Scale bar: 10 μm (for all NFAT4 experiments n = 130 (HEK293), n = 72(ORAI1-SKO), n = 82(ORAI2-SKO), n = 79(ORAI3-SKO), n = 90(ORAI1,2-DKO), n = 59(ORAI1,3-DKO), n = 71(ORAI2,3-DKO), n = 59 (ORAI-TKO); all points are single cells). Scatter plots are represented as mean ± SEM. All data in Fig. 7 was analyzed using the Kruskal–Wallis one-way ANOVA with multiple comparisons and was derived from a minimum of three independent experiments. (*p < 0.05; **p < 0.01; ***p < 0.001; ****p < 0.0001).

than those of ORAI1. From our various ORAI isoform knockout combinations, it is clear that each ORAI isoform can function alone under native conditions to support Ca$^{2+}$ signals. Nevertheless, single and double ORAI-KO have drastically altered Ca$^{2+}$ signals at all ranges of agonist concentrations (e.g., the sum of the behavior of ORAI1-SKO and ORAI2,3-DKO cells does not correspond to that of WT HEK293 cells). Further, 2-APB-mediated activation of ORAI3 is only apparent in cells lacking ORAI2 and/or ORAI1. These results strongly argue that native ORAI isoforms heteromerize with one another and that through this process, ORAI2 and ORAI3 associations with ORAI1 regulate CRAC channel activity to produce Ca$^{2+}$ signals that match the diversity of agonist strengths. Indeed, STIM1 co-expression with two different ORAI isoforms or with ORAI concatenated heterodimers generates CRAC channels with an intermediary peak current and CDI. Furthermore, our mathematical modeling lend strong support to ORAI isoform heteromerization. Our model could reproduce the Ca$^{2+}$ oscillations data obtained experimentally only when: (1) ORAI2 and ORAI3 associate preferentially with ORAI1, rather than with themselves; (2) the ORAI1,2 and ORAI1,3 heteromers allow substantially less Ca$^{2+}$ current than

does the ORAI1,1 homomer; and (3) that cells express approximately two- to fivefold more of ORAI1 proteins than ORAI2 or ORAI3 proteins. These predictions are consistent with the experimental data.

Our data supports that the native CRAC channel is constructed through heteromeric associations of ORAI isoforms, which implies that multiple ORAI heteromeric combinations with distinct biophysical properties likely adorn the surface of cells. Depending on the relative expression of each ORAI isoform, one would expect that different cell types will contain a distinct set of heteromeric CRAC channels that uniquely shape their Ca$^{2+}$ signaling profile. If the position of an ORAI isoform within the hexameric channel is not biologically relevant and only the number of each ORAI isoform matters (e.g., 131,313 is identical to 113,313), then there are 28 potential hexameric channel combinations that can be easily calculated: 3 homohexamers; 5 heterohexamers for each 2 isoform combinations (ORAI1,2; ORAI1,3 and ORAI2,3 = 15 total) and 10 heterohexamers for 3 isoform combinations. If the position of each isoform within the hexamer is important, then the number of total hexameric combinations is 92 based on Burnside's Lemma calculations with

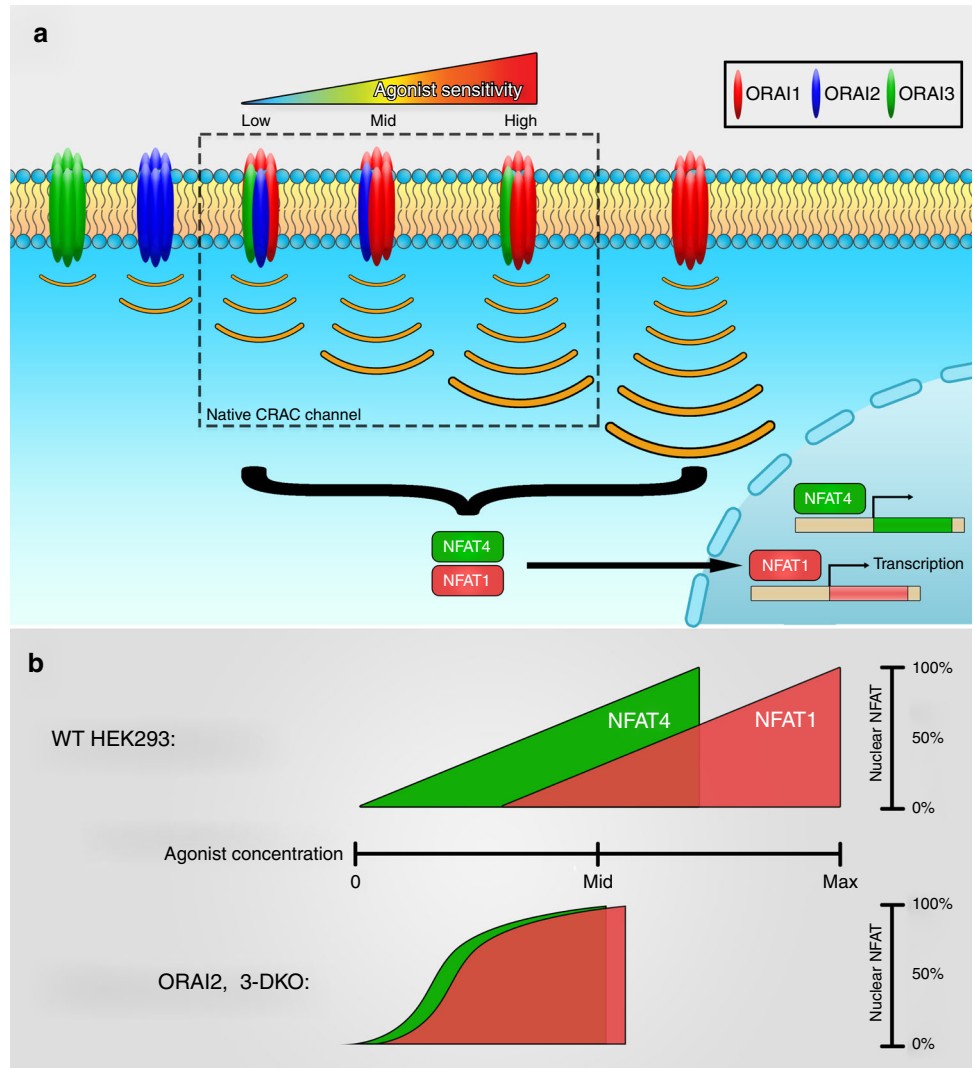

**Fig. 8 Summary of findings from the current study. a** Native CRAC channels are constructed of ORAI isoform heteromers (dashed box). Incorporation of ORAI3 and ORAI2 with ORAI1 into hexameric CRAC channels fine-tunes SOCE to gradually match the strength of agonist stimulation. CRAC channels containing ORAI2 or ORAI3 or both will have weaker activity but will more readily activate through preferential interactions with STIM1 at low levels of store depletion. **b** In wild-type cells, ORAI2 and ORAI3 expand the range of sensitivity of receptor-activated $Ca^{2+}$ signals to induce differential NFAT isoform nuclear translocation. In their absence, both NFAT1 and NFAT4 are substantially induced in response to low concentrations of agonist. Although native ORAI2 and ORAI3 can mediate SOCE when other ORAI isoforms are absent, they are incapable of supporting NFAT nuclear translocation. Whether ORAI isoform homohexamers exist under native conditions is uncertain. Our data suggest that ORAI3 and ORAI2 isoforms are unlikely to function independently of ORAI1 under native conditions. However, the precise stoichiometry or stoichiometries of native CRAC channels is likely cell type-specific and requires further investigations.

3 homohexamers, 33 heterohexamers for 2 isoform combinations (11 for each 2 ORAI isoform combinations), and 56 heterohexamers for 3 isoform combinations. (see methods and Supplementary Fig. 12 for calculations). The expression of concatenated heterohexamers with defined numbers of each ORAI isoform in ORAI-TKO cells can help determine whether the position of each isoform within the hexamer is critical for its properties or only the number of subunits of that isoform matters. Whether native CRAC channels are combinations ORAI1/ORAI3 and ORAI1/ORAI2 heterohexamers, or combinations of ORAI1/ORAI2/ORAI3 heterohexamers, or both is an important question that remains to be answered experimentally. Obviously, the exact nature and the number of different cellular combinations of ORAI heterohexamers under native conditions is difficult to predict mathematically and requires laborious experimentation to resolve. Nevertheless, our work supports the

model depicted in Fig. 8, whereby incorporation of ORAI3 and ORAI2 with ORAI1 fine-tunes CRAC channel activity in a graded manner to match the $Ca^{2+}$ responses to the wide range of agonist concentrations, and selective activation of specific NFAT isoforms and corresponding physiological functions.

Indeed, our data show that ORAI2 and ORAI3 allow differential regulation of NFAT1 and NFAT4 nuclear translocation in response to different concentrations of agonist when ORAI1 is present. However, ORAI2 and ORAI3 are incapable of mediating NFAT translocation in the absence of ORAI1. We show that NFAT1 and NFAT4 are selectively activated by $Ca^{2+}$ entering the cell through ORAI1. Importantly, despite $Ca^{2+}$ oscillations being preserved in ORAI1-SKO cells and being indistinguishable from those of WT HEK293 cells, neither NFAT1 nor NFAT4 are translocated into the nucleus, suggesting that NFAT isoforms are not decoders of the frequency of $Ca^{2+}$ oscillations as suggested

earlier[14,18,19,35], and that $Ca^{2+}$ oscillations are not critical for NFAT nuclear translocation and are likely an epiphenomenon of the stochastic activity of $IP_3R$ channel clusters that is more easily resolved with fluorescent dyes under low concentrations of agonist. Rather, our data suggests that the concentration of the $Ca^{2+}$ microdomain closely associated with ORAI1 is the driver of NFAT1 and NFAT4 activities. Whether ORAI2 and ORAI3 can, independently of ORAI1, couple to activation of other pathways and transcription factors is an interesting question that requires future investigations. Recent reports showed that ORAI2 and ORAI3 are negative regulators of ORAI1-mediated SOCE activated by maximal store depletion in T-lymphocytes, macrophages, mast cells and enamel cell lines[29,36,37], in agreement with our findings. Previous studies implicated ORAI1/ORAI3 heteromeric channels in $Ca^{2+}$ entry pathways activated in the absence of robust store depletion[38–43], which would be compatible with the basal interaction of ORAI3 with STIM1 and heightened sensitivity of ORAI3 activation by low concentrations of agonists that might not cause measurable store depletion.

Our findings have broader physiological and pathophysiological implications for all cell types where the levels of ORAI3 and ORAI2 relative to ORAI1 can alter the magnitude of SOCE and downstream signaling. For instance, we previously showed that upregulated ORAI3 contributes to native SOCE and tumorigenesis in a specific subset of breast cancer cells[44–46]. In this case, ORAI3 upregulation might offer the cancer cells a survival advantage by contributing to SOCE-mediated signaling, while limiting SOCE-mediated toxicity. A recent study showed that ORAI2 is a significant contributor to SOCE in neurons and a mediator of $Ca^{2+}$ overload during ischemia[47]. Future studies into the relative abundance of ORAI isoforms and the stoichiometry or stoichiometries of CRAC channels in different primary cell types will generate useful information into the mechanisms of CRAC channel contributions to cell physiology and disease.

## Methods

**Cell culture and transfection**. Parental HEK293 cells were obtained from ATCC (Catalog # CRL-1573) and maintained in high glucose (4.5 g/L) Dulbecco's modified Eagle's medium (DMEM) supplemented with 1× Antibiotic–Antimycotic solution and 10% heat-inactivated fetal bovine serum. Cells were maintained in a humidified $CO_2$ incubator under standard cell culture conditions (5% $CO_2$; 95% air; 37 °C). Transfections were performed using the Amaxa Nucleofector II (program Q-001; Kit V) per the manufacturer instructions. In brief $1 \times 10^6$ cells were resuspended in solution V and the cDNA plasmid added to the cell suspension. Once mixed cells were placed in a standard cuvette, transfected, and resuspended in complete DMEM media. Once resuspended, cells were transferred to 25 mm glass coverslips and returned to the incubator for 24 h prior to experimentation.

**Genetic knockout of ORAI in HEK293 cells using CRISPR/Cas9**. Two methods were used to generate ORAI-KO HEK293 cell lines. The first method utilized a single ORAI specific gRNA subcloned into the lentiCRISPR v2 vector (Addgene, Plasmid #52961) to generate insertion/deletions within the ORAI gene. Wild-type HEK293 were transfected with the corresponding ORAI1 or ORAI3 targeting vector using nucleofection as described above. Two days after transfection, puromycin (2 μg/ml) (Gemini Bio Products, West Sacramento, CA) was added to the medium to select for cells expressing the lentiCRISPR v2 vector. Six days after puromycin selection, cells were collected and seeded at a density of 1 cell/well into 96-well plates. Once colonies had formed clones were screened using the Guide-it Mutation Detection Kit (Clontech Laboratories, 631443) and genetic knockout confirmed by Sanger sequencing. This method was used to generate both ORAI1-SKO clones (#1 and #6) and ORAI3-SKO clone #25. All gRNA sequences used to generate ORAI-KO cell lines can be found in Supplementary Table 4.

The second method was later adapted to achieve genomic deletion of ORAI2 and ORAI3 in the remaining set of SKO, DKO, and TKO clones. We subcloned two ORAI2 or two ORAI3 specific gRNAs flanking the entire gene into two fluorescent vectors (pSpCas9(BB)-2A-GFP and pU6-(BbsI)_CBh-Cas9-T2A-mCherry: Addgene). Twenty-four hours after transfection, single cells with high expression of GFP and mCherry were sorted into 96-well plates (FACS Aria SORP high-performance cell sorter). Cells were then maintained in complete medium until colonies began to form. Visible colonies were collected, DNA extracted, and screened using specific primers designed to resolve a WT or knockout PCR product. Positive clones were chosen when only the knockout band was seen with

no product in the WT reaction. The complete list of primers and gRNAs can be found can be found in Supplementary Tables 2 and 4.

**Plasmids and siRNA**. Novel constructs including the ORAI2,3 concatenated heterodimer, all tk-driven ORAI1/2/3 constructs, CRISPR gRNAs, and ORAI dominant negative pore mutants were generated using primers compatible with the in-fusion cloning system (see Supplementary Table 2). Resulting plasmids were sequenced using selective primers to ensure proper cDNA insertion or mutation. All other plasmids were either generous gifts from the mentioned collaborators or purchased directly from Addgene. The siRNA sequences used to knockdown ORAI1 are listed in Supplementary Table 3.

**Fluorescence imaging**. Twenty-four hours before cell imaging, $1.5 \times 10^5$ cells were seeded onto 25 mm glass coverslips and allowed to attach. Once attached, cells were incubated in complete DMEM containing the ratiometric $Ca^{2+}$ indicator Fura2 AM (2 μM) under standard cell culture conditions. After 30 min, excess Fura2 AM was washed away and coverslips were mounted in an Attofluor cell chamber. Prior to recording, cells were washed three times with 2 mM $Ca^{2+}$ Hepes-buffered salt solution (120 mM NaCl, 5.4 mM KCl, 0.8 mM $MgCl_2$, 20 mM Hepes, 10 mM Glucose adjusted to pH 7.4 with NaOH) supplemented with 2 mM $Ca^{2+}$, as described previously[48,49]. Fura2 AM was alternately excited at 340 and 380 nm using a fast shutter wheel (Sutter Instruments) and the resulting 510 nm emission was collected from individual cells on a pixel by pixel basis through a ×20 fluorescence objective using a Hamamatsu Flash 4 camera and processed using Leica Application Suite X software.

Whole-cell and nuclear NFAT1/4-GFP (Addgene) fluorescence was measured using an ×40 oil-immersion objective paired with a Hamamatsu Flash 4 camera and processed using the Leica Application Suite X software. NFAT1/4-GFP was excited using a 488 nm fast wheel filter and emission spectra selectively captured through a GFP filter cube. Using the equation

$$\text{NFAT translocation} = \left( \frac{\text{Nuclear } F_{510}}{\text{Total cell } F_{510}} \right) \qquad (1)$$

NFAT nuclear translocation was quantified as a function of time in response to 10 μM Cch as previously described[50]. All other fluorophores were imaged using the same system and corresponding excitation filters.

**Förster resonance energy transfer microscopy**. Images were collected at room temperature using a Leica DMI 6000B inverted automated fluorescence microscope equipped with a ×40 oil-emersion objective lens (N.A.1.35; Leica). At each time point three images were sequentially collected allowing for simultaneous quantification of CFP, YFP, and FRET fluorescence, respectively, using SlideBook 6.0 software (Intelligent Imaging Innovations). Exposure times were fixed across experiments at 250 ms (YFP), 1000 ms (CFP), and 1000 ms (FRET). Three channel FRET was calculated using the formula

$$F_C = I_{DA} - \frac{Fd}{Dd} \times I_{DD} - \frac{Fa}{Da} \times I_{AA} \qquad (2)$$

where, $I_{DD}$, $I_{AA}$, and $I_{DA}$ were the respective fluorescent intensities of CFP, YFP, and FRET channels, respectively, after background subtraction. Calculation of E-FRET was performed as previously described using the formula[21,49]

$$E_{app} = Fc \frac{Fc}{Fc + G \times I_{DD}} \qquad (3)$$

To determine FRET between two ORAI isoforms, CFP/YFP-tagged ORAI constructs were transfected with STIM1-mCherry into ORAI-TKO HEK293. To achieve comparable levels of fluorescence for all ORAIs, 3, 4, and 5 μg of ORAI1, ORAI2, and ORAI3, respectively, were used. CFP, YFP, and FRET images were collected from cells with similar CFP/YFP ratios at baseline and 5 min after treatment with either increasing doses of Cch (10, 30, or 100 μM) or after maximal store depletion with 1 μM ionomycin.

Similar constructs and methods were used to determine FRET between CFP-tagged ORAI isoforms and STIM1-YFP. To achieve comparable YFP/CFP ratios, CFP-ORAI1 (2 μg), CFP-ORAI2 (2 μg), and CFP-ORAI3 (6.5 μg) were co-expressed with (1.5 μg) of STIM1-YFP. In these experiments, FRET was measured every 20 s for 10 min.

**Western blot analysis**. Cells were counted using trypan blue exclusion then washed with ice-cold phosphate-buffered saline before lysis in ice-cold RIPA buffer (150 mM NaCl, 1.0% IGEPAL CA-630, 0.5% sodium deoxycholate, 0.1% SDS, 50 mM Tris pH 8.0; Sigma) supplemented with protease and phosphatase inhibitor (Halt; Thermo Scientific) for 30 min. Lysates were centrifuged at $15,000 \times g$ for 12 min at 4 °C and clarified supernatant collected for measurement of protein concentration. Using the Pierce BCA assay, protein extracts were quantified prior being loaded into a NuPAGE 4–12% Bis-Tris precast protein gels. Migrated proteins was transferred to polyvinylidene difluoride membranes using the transblot turbo system (Biorad). Membranes were then blocked with LI-COR Tris-buffered saline (TBS) buffer for 1 hr at room temperature and incubated with primary antibody overnight at 4 °C. Probed membranes were washed with TBS with Tween

(TBST) then probed while shaking with secondary antibodies for 1 h at room temperature. Resulting blots were imaged and quantified using a LI-COR odyssey imaging system running Image Studio software (LI-COR).

**cDNA generation and quantitative PCR**. HEK293 Cells ($4 \times 10^6$) were lysed in TRIzol (Invitrogen) and total RNA extracted according to the manufacturer protocol. Isolated RNA was quantified using a nanodrop 2000 spectrophotometer (Thermo Scientific) and 1 μg of RNA used to generate cDNA using the High-Capacity cDNA Reverse Transcription Kit according to the manufacturer's protocol (Applied Biosystems). Total cDNA (1 μL) was then added to SYBR Green qPCR Master Mix (Applied Biosystems) and the corresponding primers added resulting in a 10 μL reaction. Samples were then loaded into a 96-well plate and data from the resulting reaction was collected and quantified using the Quant-Studio 3 real-time PCR system (Applied Biosystems). The PCR protocol for all reactions began with an initial activation 2 min 50 °C step followed by a 95 °C 2 min melt step. The initial steps were followed by 40 cycles that began at 95 °C for 15 s followed by 15 s at 54.3 °C and 72 °C for 30 s. After completion of 40 cycles, a standard melt curve was generated to ensure primer specificity. All primer sequences used are provided within Supplementary Table 2. Analysis of target and control samples was carried out using the comparative Ct method. Experimental samples were normalized to the reference gene glyceraldehyde-3-phosphate dehydrogenase. All data were generated in technical and biological triplicates to ensure reproducibility.

**Analysis of Ca$^{2+}$ oscillations**. The individual $F_{340}/F_{380}$ for each cell as a function of time was plotted using GraphPad Prism 8. All oscillations resulting from experiments were then transferred to layouts and oscillations per 15 min were manually counted. Cells were divided into three groups as follows: (1) the first group (oscillating cells) represents cells that display regenerative oscillations for the duration of the experiment, where each oscillation returns to baseline before the start of the next oscillation. (2) The second group (non-responders) are cells that show either no response at all to agonist stimulation or showed only one initial spike and remained at baseline for the duration of the recording. (3) The third group (plateau cells) sustained a cytosolic Ca$^{2+}$ signal that was ≥25% of the initial peak, for at least 5 min after stimulation (Supplementary Fig. 11). On a coverslip by coverslip basis, cells were grouped into these three categories and a percentage of each group of cells was calculated for each individual coverslip. Within the first group of oscillating cells, the total number of oscillations/cell during 14 min were manually counted on a cell by cell basis. All manual counts were independently recorded by two individuals to ensure accuracy of counts.

**Patch-clamp electrophysiology**. For experiments assessing ORAI fast CDI, HEK293 ORAI-TKO cells were transfected with YFP-STIM1 (4 μg) and 1 μg of one either CFP-tagged ORAI isoform or 1 μg of tdTomato-tagged ORAI heterodimer. For experiments with co-expression of two ORAI isoforms, 0.5 μg of each CFP-ORAI isoform was co-transfected with 4 μg of YFP-STIM1.

Cells were seeded onto 30 mm glass coverslips 24 h before recordings. Traditional patch-clamp electrophysiological recordings were performed using an Axopatch 200B and Digidata 1440A (Molecular Devices). Pipettes were pulled from borosilicate glass capillaries (World Precision Instruments) with a P-1000 Flaming/Brown micropipette puller (Sutter Instrument Company) and polished using a DMF1000 (World Precision Instruments). The resistances of all filled pipettes ranged from 2 to 4 MΩ. Recordings were performed only on cells forming tight seals (>16 GΩ) and only when the whole-cell configuration generated less than 8 MΩ values of series resistances. As a precaution, 8 mM MgCl$_2$ was included in the pipette solution to inhibit TRPM7 currents. The Clampex 10.3 software was used for data collection and Clampfit 10.3 software (Molecular Devices) was used for data analysis.

All CDI recordings were measured from ORAI-TKO cells co-expressing STIM1 and ORAI constructs driven by CMV promotors. STIM1 and ORAI constructs used for CDI were transfected at a 4 μg (STIM1) to 1 μg (ORAI) ratio. For ORAI isoform co-transfections, 0.5 μg of each ORAI cDNA was transfected totaling 1 μg ORAI cDNA per condition. Expression of both STIM1 and ORAI constructs was verified by fluorescence microscopy. Immediately after break-in and before CRAC currents had developed, families of 150 ms voltage steps (from +30 mV holding potential to −120, −100, −80, and −60 mV) were run with 2 s intervals between steps. At the beginning of each pulse, a 2.5 ms voltage step to 0 mV was used to eliminate residual capacitive artifacts. The first voltage step from each recording was subtracted from the resulting trace as leak current. Next, a 250 ms voltage ramp from +100 to −140 mV was administered every 2 s until CRAC currents reached steady-state level (usually within 150–300 s). Once steady state was achieved, CDI was determined by performing a second round of voltage steps, using the same protocol as described above. CDI was quantified as the remaining current measured at 146 ms from the peak current at the beginning of the pulse. The bath solution contains 115 mM Na-methanesulfonate, 10 mM CsCl, 1.2 mM MgSO$_4$, 10 mM Hepes, 20 mM CaCl$_2$, and 10 mM glucose (pH 7.4 with NaOH). The pipette solution contains 135 mM Cs-methanesulfonate, 10 mM EGTA, 8 mM MgCl$_2$, and 10 mM Hepes (pH 7.2 with CsOH).

Recordings of native CRAC currents were performed on WT HEK293, ORAI-SKO, ORAI-DKO, and ORAI-TKO cell lines. Cells were maintained at a +30 mV holding potential during experiments and subjected to reverse voltage ramps from +100 to −140 mV lasting 250 ms at 2 s intervals. Immediately after establishing the whole-cell configuration, a 30 s DVF pulse (before current development in response to store depletion by 1,2-bis(o-aminophenoxy)ethane-N,N,N′,N′-tetraacetic acid (BAPTA)) was applied to gauge maximal background current. The $I/V$ curves obtained in DVF bath solutions, representing the maximal background current, are averaged and then subtracted from the average $I/V$ of BAPTA- or 2-APB-activated Na$^+$ currents obtained in DVF bath solutions when the current is maximal. These subtracted $I/V$ curves are represented as independent $I/V$ curves in all figures. The bath solution contains 115 mM Na-methanesulfonate, 10 mM CsCl, 1.2 mM MgSO$_4$, 10 mM Hepes, 20 mM CaCl$_2$, and 10 mM glucose (pH 7.4 with NaOH). The pipette solution contains 115 mM Cs-methanesulfonate, 20 mM Cs-BAPTA, 8 mM MgCl$_2$, and 10 mM Hepes (pH 7.2 with CsOH). The DVF solution contains 155 mM Na-methanesulfonate, 10 mM HEDTA, 1 mM EDTA, and 10 mM Hepes (pH 7.4 with NaOH).

**Mathematical modeling**. The Ca$^{2+}$ dynamics model is an adapted version of that in ref. [22]. In brief, the cell is assumed to be well-mixed and the IP$_3$Rs are modeled by a deterministic model[51] that was originally based on the stochastic model of ref. [52]. STIM are not modeled explicitly and thus ORAI activation is an algebraic function of ER [Ca$^{2+}$].

The major difference from ref. [22] is the modeling of the Ca$^{2+}$ influx across the PM, which we do by a simple ORAI binding model. ORAI1 ($O_1$), ORAI2 ($O_2$), and ORAI3 ($O_3$) bind to form dimers according to the reactions

$$O_1 + O_1 \overset{K_{11}}{\leftrightarrow} O_{11}, O_2 + O_2 \overset{K_{22}}{\leftrightarrow} O_{22}, \tag{4}$$

$$O_1 + O_2 \overset{K_{12}}{\leftrightarrow} O_{12}, O_2 + O_3 \overset{K_{23}}{\leftrightarrow} O_{23}, \tag{5}$$

$$O_1 + O_3 \overset{K_{13}}{\leftrightarrow} O_{13}, O_3 + O_3 \overset{K_{33}}{\leftrightarrow} O_{33}, \tag{6}$$

where we assume that all these reactions are at equilibrium. Subsequent combination of these dimers into hexamers (as is believed to be the actual case) is not modeled here, due to the greatly additional combinatorial complexity (with either 56 or 28 distinct hexamers, depending on the exact assumptions) that would be required. The available data are not sufficient to justify such an increase in model complexity.

Assume that the unstimulated cell starts with initial amounts, $I_1$, $I_2$, and $I_3$ of ORAI1, ORAI2, and ORAI3, respectively. We set $I_1 = 1$, $I_2 = 0.2$, and $I_3 = 0.2$. Only the relative values matter. $I_2$ and $I_3$ can be increased to around 0.5 without making any significant qualitative difference.

We have the equilibrium equations

$$O_{11} = K_{11}O_1^2, \tag{7}$$

$$O_{12} = K_{12}O_1O_2, \tag{8}$$

$$O_{13} = K_{13}O_1O_3, \tag{9}$$

$$O_{22} = K_{22}O_2^2, \tag{10}$$

$$O_{23} = K_{23}O_2O_3, \tag{11}$$

$$O_{33} = K_{33}O_3^2. \tag{12}$$

Furthermore, conservation of ORAI gives

$$2O_{11} + O_{12} + O_{13} + O_1 = I_1, \tag{13}$$

$$O_{12} + 2O_{22} + O_{23} + O_2 = I_2, \tag{14}$$

$$O_{13} + O_{23} + 2O_{33} + O_3 = I_3 \tag{15}$$

and thus, combining with the equilibrium equations, we get

$$2K_{11}O_1^2 + K_{12}O_1O_2 + K_{13}O_1O_3 + O_1 = I_1, \tag{16}$$

$$K_{12}O_1O_2 + 2K_{22}O_2^2 + K_{23}O_2O_3 + O_2 = I_2, \tag{17}$$

$$K_{13}O_1O_3 + K_{23}O_2O_3 + 2K_{33}O_3^2 + O_3 = I_3. \tag{18}$$

Finally, we assume that each $O_{ij}$ has a conductance of $\alpha_{ij}$ and so the total Ca$^{2+}$ influx through the population of dimers, $J_{in}$, will be given by

$$J_{in} = \alpha_{11}O_{11} + \alpha_{12}O_{12} + \alpha_{13}O_{13} + \alpha_{22}O_{22} + \alpha_{23}O_{23} + \alpha_{33}O_{33}. \tag{19}$$

Oscillation frequency in the model is determined by $J_{in}$. Thus, to get a model oscillation with the same period as that observed experimentally (e.g., Fig. 1a–l), one simply need choose $J_{in}$ appropriately. Let $d_{ij}$ denote the influx when ORAI $i$ and $j$ are present. By ensuring that the model oscillation has the desired period, we can estimate $d$ for the WT, DKO, and TKO cases. We note that these estimations

are not precise, as there is considerable variability in the experimental results. For the present model, this gives

$$d_{11} = 0.06 = \alpha_{11}O_{11}, \tag{20}$$

$$d_{12} = 0.05 = \alpha_{11}O_{11} + \alpha_{12}O_{12} + \alpha_{22}O_{22}, \tag{21}$$

$$d_{13} = 0.05 = \alpha_{11}O_{11} + \alpha_{13}O_{13} + \alpha_{33}O_{33}, \tag{22}$$

$$d_{22} = 0.002 = \alpha_{22}O_{22}, \tag{23}$$

$$d_{23} = 0.002 = \alpha_{22}O_{22} + \alpha_{23}O_{23} + \alpha_{33}O_{33}, \tag{24}$$

$$d_{33} = 0.001 = \alpha_{33}O_{33} \tag{25}$$

$$d_{123} = 0.045 = \alpha_{11}O_{11} + \alpha_{12}O_{12} + \alpha_{13}O_{13} + \alpha_{22}O_{22} + \alpha_{23}O_{23} + \alpha_{33}O_{33}, \tag{26}$$

The values of $d$ can be scaled arbitrarily; only the relative sizes matter. Furthermore, they depend on the exact details of the $Ca^{2+}$ model.

We now have ten equations ((16)–(18) and (20)–(26)) to solve for the nine unknowns $O_1$, $O_2$, $O_3$, $K_{11}$, $K_{12}$, $K_{13}$, $K_{13}$, $K_{23}$, and $K_{33}$ as functions of the $\alpha_{ij}$, and thus have an overdetermined system which, in general, will not have an exact solution.

Some of the equilibrium constants can be determined algebraically; this reduces complexity and imposes explicit constraints on some parameter values.

To find the equilibrium constants for the DKO cases: to calculate $K_{11}$, we have to solve the two equations

$$d_{11} = \alpha_{11}O_{11} = \alpha_{11}K_{11}O_1^2, \tag{27}$$

$$2O_1 + 2K_{11}O_1^2 = I_1. \tag{28}$$

We eliminate $K_{11}$ between these two equations to get

$$O_1 = \frac{I_1}{2} - \frac{d_{11}}{\alpha_{11}}, \tag{29}$$

and

$$K_{11} = \frac{d_{11}}{\alpha_{11}O_1^2}. \tag{30}$$

Hence, we require

$$I_1 > \frac{2d_{11}}{\alpha_{11}}. \tag{31}$$

The same approach gives us $K_{22}$ and $K_{33}$, with analogous constraints on $d_{11}$, $d_{22}$, $\alpha_{11}$, and $\alpha_{22}$.

The procedure is more complicated when only a single ORAI is knocked out. We do the ORAI3-SKO case as an example. In this case, we have the three equations

$$d_{12} = \alpha_{11}K_{11}O_1^2 + \alpha_{12}K_{12}O_1O_2 + \alpha_{22}K_{22}O_2^2, \tag{32}$$

$$I_1 = 2K_{11}O_1^2 + O_1 + K_{12}O_1O_2, \tag{33}$$

$$I_2 = 2K_{22}O_2^2 + O_2 + K_{12}O_1O_2 \tag{34}$$

to solve for the three unknowns, $O_1$, $O_2$, and $K_{12}$. There is now no simple algebraic solution. Instead, we take a numerical approach. We vary $K_{12}$ from, say, 1–500, and for each value we calculate $O_1$ and $O_2$ from Eqs. (17) and (18) (it is noteworthy that all the other parameters are known, and that the solution to (33) and (34) always exists, as can be seen from a graphical approach). Then we find numerically the root of (32), which gives us $K_{12}$. In this way, finding the roots of a three-dimensional nonlinear system simplifies to finding the root of a one-dimensional equation.

If no root exists, then it is not possible to solve for $K_{12}$. This entire procedure can be incorporated in a continuation algorithm so that $K_{12}$ can be found as a function of the parameters $\alpha_{ij}$; thus, the regions of parameter space for which a solution exists can be determined. For example, for the model we used here, solutions for $K_{12}$ and $K_{13}$ exist only when $\alpha_{11}/\alpha_{12} > \sim 20/3$ and $\alpha_{11}/\alpha_{13} > \sim 20/3$. This implies that the model is consistent with the data only when ORAI2 and ORAI3 inhibit ORAI1.

So far we have used Eqs. (16)–(18) and (20)–(25) to solve for the nine unknowns (assuming that a solution exists). Equation (26), however, is not yet satisfied in general. We now use these solutions as a starting point for a fit to Eqs. (16)–(18) and (20)–(26). Only a simple least-squares fit was done to obtain an estimate of the equilibrium constants. More detailed fits, using, e.g., a Markov-Chain–Monte-Carlo approach, are left for future work, but will be nontrivial, as will they will depend on an estimate in the errors in the data $d_{ij}$, which are highly model-dependent. As yet, it is not clear how to incorporate model variability into the parameter estimation method.

The result is $K_{11} = 14$, $K_{22} = 1.5$, $K_{33} = 0.93$, $K_{12} = 19$, $K_{13} = 17$, $K_{23} = 14$, all in units of 1/concentration. Again, the actual values are unimportant; only the ratios

have meaning. These values imply that ORAI2 and ORAI3 preferentially bind to ORAI1.

Typical simulations of the model are shown in Fig. 6. The parameter values used were $I_1 = 1$, $I_2 = 0.2$, $I_3 = 0.2$ (all in units of concentration), $K_{11} = 14$, $K_{22} = 1.5$, $K_{33} = 0.93$, $K_{12} = 19$, $K_{13} = 17$, $K_{23} = 14$ (all in units of 1/concentration), and $\alpha_{11} = 2$, $\alpha_{22} = 1$, $\alpha_{33} = 0.7$, $\alpha_{12} = 0.1$, $\alpha_{13} = 0.1$, $\alpha_{23} = 0.1$ (all in units of 1/time). It is noteworthy that the absolute values are not important, only the ratios of the values.

Figure 6a, b, c show model responses to increasing agonist concentrations (equivalent to increasing values of $p$) for the WT cells and for a selection of ORAI knockouts. In every case, an increase in agonist causes an increase in oscillation frequency, a slightly raised baseline to the oscillations, and, eventually, termination of the oscillations in a raised plateau. Panel a shows that when ORAI1 and ORAI2 are knocked out, the oscillation frequency decreases dramatically, but that ORAI3 is still capable of mediating sustained oscillations. Panel b shows that the ORAI1-SKO case has greatly decreased oscillation frequency, while the ORAI2-SKO case has a frequency only slightly decreased from the WT case in panel a. Panel c shows that when all the ORAI are knocked out, oscillations can still be sustained for a small number of peaks, but that when only ORAI2 and ORAI3 are knocked out, oscillation frequency increases from WT, and the plateau happens at lower agonist concentrations.

Panel d shows the model simulation of experiments in which thapsigargin was applied to the cell in the absence of external $Ca^{2+}$, and then, once the cell was emptied of $Ca^{2+}$, external $Ca^{2+}$ was reapplied. The resultant $Ca^{2+}$ influx is a measure of the CRAC current. The ORAI1,2-DKO current (which is current only through ORAI3) is extremely small, but is still large enough to support sustained oscillations, as shown in panel a. Conversely, the ORAI2,3-DKO current is much larger than the WT current, which implies that there is less $Ca^{2+}$ influx when ORAI2 and ORAI3 are present also.

Taken together, these model results suggest that (1) ORAI2 and ORAI3 inhibit $Ca^{2+}$ influx through ORAI1, (2) very small $Ca^{2+}$ entry through ORAI3 channels is sufficient to support $Ca^{2+}$ oscillations.

**Combinatorics of ORAI heterohexamers.** According to Burnside's Lemma, the number of distinct three colorings of the six cycle is

$$\frac{1}{12}[IC(\pi_0) + \cdots + IC(\pi_5) + IC(r_1) + \cdots IC(r_6)] \tag{35}$$

where $\pi_0, \ldots, \pi_5$ are the six rotational symmetries of the six cycle, i.e., $\pi_j$ is rotation by $j * 60°$; $r_1, \ldots, r_6$ are the six reflective symmetries of the hexagon; and $IC(\tau)$ is the number of invariant colorings under permutation $\tau$ (either a rotation or a reflection; Supplementary Fig. 12). It can be shown that

$$IC(\pi_0) = 3^3, \tag{36}$$

$$IC(\pi_1) = IC(\pi_5) = 3^1, \tag{37}$$

$$IC(\pi_2) = IC(\pi_4) = 3^2, \tag{38}$$

$$IC(\pi_3) = 3^3, \tag{39}$$

$$IC(r_1) = IC(r_2) = IC(r_3) = 3^3, \tag{40}$$

$$IC(r_4) = IC(r_5) = IC(r_6) = 3^4. \tag{41}$$

It follows that the number of distinct three colorings is

$$\frac{1}{12}\left[3^6 + 2*3^1 + 2*3^2 + 3^3 + 3*3^3 + 3*3^4\right] = 92. \tag{42}$$

Replacing base 3 in the above expression with either 1 or 2 yields the number of distinct one or two colorings for a given single color or pair of colors, respectively, i.e.,

$$\frac{1}{12}\left[1^6 + 2*1^1 + 2*1^2 + 1^3 + 3*1^3 + 3*1^4\right] = 1, \tag{43}$$

$$\frac{1}{12}\left[2^6 + 2*2^1 + 2*2^2 + 2^3 + 3*2^3 + 3*2^4\right] = 13. \tag{44}$$

Thus, by the inclusion/exclusion principle the total number of three colorings that use all three colors is: $92 - (3 \times 13) + (3 \times 1) = 56$.

**Statistics.** Statistical tests were performed using GraphPad Prism 8 and Origin 9.0 software. When comparing two groups the Student's $t$-test was used. If greater than two groups were compared, then One-way analysis of variance was used. For all results with normally distributed data, parametric statistical tests were used. When data were not normally distributed, non-parametric statistical tests were used. Data normality was determined using Prism 8.0. All data are represented as the mean and error reported as ±SEM. Throughout the manuscript *, **, ***, and **** indicate $p$-values of <0.05, <0.01, <0.001, and <0.0001, respectively. All data with $p < 0.05$ was deemed significant; "ns" = nonsignificant. The exact $p$-values for

each statistical comparison performed in the study are provided as an excel file (Supplementary Data 2).

**Reporting summary**. Further information on research design is available in the Nature Research Reporting Summary linked to this article.

## Data availability

Data supporting the findings of this manuscript are available from the corresponding author upon reasonable request. A reporting summary for this Article is available as a Supplementary Information file. This source data file includes all data points and uncropped gels for Figs. 1f, m–p, x, 2i–p, r–v, 3b–d, h–k, m–q, 4d, e, i, j, n, o, 5e–m, 6a–d, and 7b, c, e, f, i, j, l, m, and Supplementary Figs. 1a–i, 2a, 3c, f, i, 4c, f, i, l, 5e, 6a–e, 7a–c, 8c, f, i, 9c, f, i, 10a–d, and 11.

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

## Acknowledgements

We acknowledge Dr Barry Tesman (Dickinson College, Carlisle, PA) for providing assistance related to the mathematical theory of graph colorings and Dr Tamas Balla (NIH) for the gift of the Thymidine kinase (tk) promoter. Our study was supported by the National Heart, Lung, and Blood Institute (R01-HL123364, R01-HL097111, and R35-HL150778 to M.T.), and by National Institute of General Medical Sciences (1R35 GM131916 to D.L.G.).

## Author contributions

R.E.Y. and S.M.E. contributed to the design of experiments, performed most experiments, including Westerns, real-time PCR experiments, imaging experiments, $Ca^{2+}$ oscillations assays and NFAT nuclear translocation assays, validated CRISPR/Cas9 clones, analyzed the data, generated figures, wrote the first draft of the manuscript, and contributed to the editing and writing of the final version of the manuscript. X.Z. performed the patch-clamp electrophysiology experiments. P.X. generated CRISPR/Cas9 clones, and designed and performed cloning and molecular biology experiments. M.T.J. performed Westerns. A.F. helped with the $Ca^{2+}$ oscillations assays. V.W. performed combinatorics calculations of ORAI hexamers and provided critical input on statistical analysis. N.H. contributed to experimental design and editing of manuscript. D.I.Y. contributed to the design of experiments, provided critical input on the mathematical model, and provided the $IP_3R$ CRISPR/Cas9-knockout HEK293 cells and the antibodies against $IP_3R$ isoforms. J.S. devised and generated the mathematical model of $Ca^{2+}$ oscillations. D.L.G. contributed to experimental design and editing of manuscript, and provided critical constructs and reagents. M.T. conceived the hypothesis, designed the experiments, supervised the research, interpreted the data, and wrote the manuscript.

## Competing interests

The authors declare no competing interests.
