## [Peer Review File · Nature Communications]

Reviewers' comments:

Reviewer #1 (Remarks to the Author):

The manuscript "The Native ORAI Channel Trio Underlies the Diversity of Ca²⁺ Signaling Events" by Ryan E. Yoast, Scott M. Emrich et al. presents important insights into the significance of ORAI2 and ORAI3 for the molecular choreography of the CRAC channel complex. While ORAI1 has been the focus of most studies, the exact role of its two isoforms has remained largely unresolved to this date.

General impression on the manuscript:

The manuscript is arranged in a nice fashion and all experiments are presented with a clear train of thought. A wide spectrum of different methods including cell culture, molecular cloning, CRISPR/Cas9, quantitative PCR, Ca²⁺ imaging, FRET microscopy, biochemistry, electrophysiology and mathematical modeling enables the construction of a native model of a heteromultimeric CRAC channel complex composed of all three ORAI protein isoforms.

The authors reveal a specific correlation between ORAI2/ORAI3 and cellular Ca²⁺ oscillations. While these do not depend on the presence of ORAI1, the latter is indispensable for robust SOCE and sustained Ca²⁺ plateau signals. ORAI2/ORAI3 additionally heteromerize with and negatively regulate ORAI1 by strengthening Ca²⁺-dependent inactivation (CDI). Finally, the nuclear factor of activated T-cells (NFAT) family requires ORAI1 for activation with ORAI2/ORAI3 alone not being sufficient. However, NFAT isoforms require varying activation signals mainly differing in the amount of Ca²⁺ that is required. Through their negative regulation of ORAI1, ORAI2/ORAI3 enable the differential activation of NFAT isoforms in response to distinct agonists and cytosolic Ca²⁺ concentrations. With this study, the authors thus provide another key step towards elucidating the molecular choreography of the native CRAC channel complex and the diverse Ca²⁺ signaling events supported by it.

The reported results are interesting. However, the authors should address the following points to strengthen the manuscript.

Major comments:

Some implications go into the model which should be supported experimentally:

- (i) The implication that both ORAI2 as well as ORAI3 heteromerize preferentially rather than forming homomeric assemblies needs to be proven experimentally.
- (ii) The stronger interaction of ORAI2/3 with STIM1 compared to ORAI1 is suggested to result in Ca²⁺ entry at low- and mid-range agonist concentrations which only slightly deplete Ca²⁺ from the ER store.
- (iii) The impact of CDI would be interesting to see with ORAI channels devoid of it.
- (iv) Low agonist concentrations have been recently shown to mediate STIM1 activation via STIM2. Could this play here a role as well, as STIM2 is assumedly not knocked-out?

Additional points:

- The authors suggest that native expression levels will be achieved by using the weak tk promoter instead of the strong CMV-promoter (line 258), however, there is no experimental proof of that.
- In Supplementary Figure 1a isoform expression in ORAI1 knock-out cells is shown. The image is not explained in detail and one can just estimate which bands can be seen on this image. Should the prominent band below 25 kDa correspond to ORAI1? If so, the authors need to state why there is a difference between observed sizes on western blot and the expected, calculated size of endogenous ORAI1 proteins of ~33kDa. Moreover, there are two additional bands visible right below 37kDa – there is no explanation given in the manuscript what these bands represent.
- CRISPR/Cas9 method used: The authors state that they used two different methods for knock-out construction. First, it is not clear to me which cell clones (that were used within the manuscript) derived from which method. Second, and even more important: when using the first

method using one gRNA targeting ORAI1, some points need to be clarified:

\ gRNA is targeting ORAI1 at ~amino acid 145. Evidence is given by literature that the DSB usually occurs 100bp upstream or downstream of the gRNA binding site. Still, designing the gRNA this way, there will be ~110 amino acids expressed of the native ORAI1 protein. As the ORAI1 antibody used for experiments binds to a peptide at the very end of the ORAI1 protein (288-301) the authors can not exclude that there is still a functional part of ORAI1 left.

\ In conjunction with the previous point, some of the cell lines used in this paper (ORAI3-SKO Clone #25, ORAI2,3-DKO Clones #9 and #22) also allow the translation of small residual ORAI3 protein fragments that could potentially interfere with other ORAI isoforms or with ORAI3 artificially introduced by transfection.

\ Supplementary Table 2: ORAI1 gRNA does not match the sequence given in the methods section. It is extended by CACG-, a sequence that does not bind ORAI1.

\ Supplementary Figure 2b/c: In Supplementary Figure 2b and 2c, the PCR results of genomic DNA are intended to show large genomic deletions of ORAI2 to validate gene knockout. However, several issues with these figures remain to be clarified: (1) it is unclear whether the WT and KO boxes belong to the same gel or show different gels, (2) WT seems to indicate ORAI1-SKO cells, probably confusing the reader, (3) it is not explained why more than one band appears for clones #19 and #22, and (4) along with the issue raised in (1), the presence of size-marker labeling would make it easier to interpret the results.

- There are several typing-errors throughout the text (line 121, 132, 159, 392, ..)
- Methods section – FRET microscopy: it is stated that a STIM1-mCherry plasmid was used for transfection – but this does not show up anywhere in the figures.
- Figure 2o: whole blots have to be shown and the question arises how the size of ORAI1 shown here (37kDa) can be correlated to the different sizes shown in Supplementary Figure 1a.
- ORAI isoform heteromerization is mimicked by the use of concatenated dimers. In line 312/313 the authors claim that peak currents of dimer constructs are smaller due to the lower expression of these constructs. Yet, there are no experiments presented where expression levels and correct localization of dimer constructs are shown.
- The authors make excessive use of subfigures, a number of which could easily be merged in some form (e.g. Figure 1 r-w). The figures would also benefit from the use of visual cues to indicate/group related subfigures. Finally, the references for significance testing should be better indicated in all cases, as simple asterisks (*) do not fully clarify that comparisons are only related to wild type cells (cf. comment to Figure 1x below).
- The number of cells used for the calculation of mean values and their corresponding SEM are not indicated in many applicable subfigures across the manuscript.
- It is claimed that ORAI3-SKO cells show “increased SOCE in response to 2 μ M thapsigargin” and “increased oscillatory frequency triggered by 10 μ M carbachol” when compared to wild type cells, although I couldn’t see the underlying statistical analysis.
- With reference to Figure 1x, the authors report that “ORAI2,3-DKO cells show an increase in SOCE in response to thapsigargin” when compared to wild type cells and that “this increase was significantly bigger than that of ORAI2-SKO and ORAI3-SKO cells” (l 175). However, all significance testing in Figure 1x relates to wild type cells (as is also indicated in the corresponding figure legend).
- It would be interesting to see respective ORAI localization images when expressed in ORAI-TKO cells.
- Line 360: Mathematical model taking only “ORAI dimer formation” into account: I guess the meaning is that only two ORAI isoforms contribute to the hexameric ORAI channel.

Reviewer #2 (Remarks to the Author):

In the current study, Yoast et al. perform a comprehensive analysis of the contribution of each of the three ORAI proteins in HEK cells for calcium homeostasis following activation of agonist/receptor-operated and store-operated Ca²⁺ entry pathways. They generate and analyse seven different genotypes after generation single, double- and triple-compound ORAI1 knockout alleles in HEK cells covering all possible combinations of ORAI subtype deletions to study the distinctive role of individual ORAI proteins for calcium entry and/or oscillations generated by different strengths of receptor stimulation by comparison with wildtype HEK cells. The authors study heteromultimerization in this cellular system using FRET analysis upon expression of fluorescently tagged ORAI1 proteins or expression of individual ORAI proteins with established pore mutations. The readout for a cellular function that can be attributed to the distinct calcium signaling pattern conveyed by individual ORAI proteins is nuclear translocation of transcription factor NFAT1 and NFAT4. Using this approach, the authors elaborate an essential role of ORAI1 as the key ORAI subtype mediating store-operated calcium entry as determinant of NFAT1 and NFAT4 translocation to the nucleus.

There are several issues that need to be addressed (see below), but overall this is a very carefully performed study analysing systematically and comprehensively the contribution of all ORAI isoforms in a given cell system under native expression levels which has not been demonstrated before in the literature. Overall the results are convincing and support the conclusions.

Comments.

1. The generation of the individual and compound ORAI knockout cell lines seems to be carefully performed, but the documentation of the molecular characterization needs to be documented in more detail in the supplementary figures where space is not limited. The authors should provide the sequence (deletions, insertions) for all individual alleles for ORAI1, ORAI2 and ORAI3 single knockouts as well as the double and triple knockout cell clones. The analysis of compensatory up- and down-regulation of STIM and IP3 receptor proteins is well performed. However, up- or down-regulation of ORAI2 and ORAI3, respectively, should be addressed in all knockout clones by qPCR. This has only been partially performed in supplementary figure 2, e.g. expression of ORAI3 should be shown in ORAI1&2 double knockout clones (# 18 and 19), ORAI2 expression in ORAI1/3 double knockout (clone #48, clone #3) and similarly in ORAI2 SKO and ORAI3 SKO clones. Also, supplementary figure 2b and c are difficult to understand. They should be correlated with the documentation of the gene edited sequences of the individual clones in more detail.

As no specific antibodies are available for Orai2 and Orai3, respectively, the authors should compare the expression levels of ORAI1, 2 and 3 transcripts in their cell system, i.e. HEK cells (may be they have RNA seq data for this).

2. On page 6, line 164 the authors state that ORAI3-SKO cells show increase in SOCE. This does not fit to the results shown in figure 1t.

3. Page 6, line 176 the authors state that Thapsigargin-evoked SOCE is increased in ORAI2/3 DKO cells is bigger than that of ORAI2-SKO and ORAI3-SKO. This does not fit to the data shown in Figure 1x. Was multiple comparison analyzed by ANOVA?

4. The authors should define more precisely what qualifies cellular response into the category "plateau cells". The authors only state that the intracellular calcium level should be above baseline for longer than five minutes. What is above baseline? more than 5% or 2% or what is the cut off?

5. On page 8, line 269 the authors state that TK promotor driven ORAI3 expression leads to significant SOCE. In figure 3a (green average trace) the Ca²⁺ entry seems not to be significantly larger than in the ORAI1 TKO cells. Please comment and provide quantitative comparison.

6. In figure 5 the authors study ORAI1 pore mutant in ORAI1-SKO cells (A, B). The authors then proceed and study ORAI2 and ORAI3 pore mutants (C, D), respectively again in ORAI1-SKO cells. Shouldn't the ORAI2 pore mutant be studied in ORAI2-SKO and ORAI3 pore mutant in ORAI3-SKO to study functional interaction with the other corresponding ORAI isoforms?

7. The authors make use in figure 5g-m of 2-APB which has different action on ORAI1, ORAI2 and ORAI3. Interestingly, the small but remaining current with a typical I-V curve of a CRAC channel is enhanced by 2-APB in ORAI1-SKO cells. Are ORAI2- and ORAI3-mediated currents sensitive to other typical CRAC channel blockers such as GSK7975A?

8. Page 10 line 354: line 352 the authors state that 2-APB potentiates currents in cells lacking

ORAI2 (figure 5i). This statement is not supported by the data shown.

9. In figure 7 the authors show convincingly that nuclear translocation of NFAT1 and NFAT4 is differently regulated upon deletion of ORAI1 on the one hand and ORAI2/3 inactivation on the other hand. These experiments correlate very well with the effect of the individual ORAI deletion on the extent of an amplitude of SOCE. ORAI2 and 3 are only modulating but not responsible for the amplitude of store-operated calcium entry (as opposed to ORAI1) but are key regulators of calcium oscillations, particularly at the carbachol concentrations studied (10mM); thus, the authors should aim to analyze the activation of transcription factors that do not depend on a plateau calcium elevation but are regulated by the frequency of calcium oscillations. NFkB has been reported already many years ago as such calcium dependent transcription factor (Dolmetsch et al., Nature 386: 855-8, 1997) and should be studied in the Orai Ko cells (ORAI1 vs ORAI2/3) or other transcriptional responses for which the authors could show dependence on the Ca²⁺ oscillation frequency.

Marc Freichel

Reviewer #3 (Remarks to the Author):

In this manuscript, the authors investigate the roles of Orai2 and Orai3 in Ca²⁺ entry in physiological conditions, i.e. when cells are stimulated by moderate doses of agonists. The amount of results is impressive, and the experiments are very clear. The manuscript represents a significant contribution to the understanding of Orai-mediated Ca²⁺ influx and to Ca²⁺ signalling in general.

My report concerns the modelling part.

The model is well-built and provides an interesting contribution to the message of the manuscript. It clarifies and strengthens the main messages. The model used in this study is based on a reliable model of Ca²⁺ oscillations (ref. 22) that has already been validated against experimental data. When coupled to the part describing Ca²⁺ influx via the different Orai hexamers, it leads to an impressive agreement with the experimental data presented here. Some points may require clarifications, as suggested below:

(1) How were the dij values determined? The authors write that it was done "by comparing the model output qualitatively to the experimental results". Intuitively, I would have fixed the ratios of the dij by looking at the experiments shown in Fig. 1, panels r to w. It is obviously not what has been done. Why? The qualitative procedure should be explained.

(2) In the PNAS paper (ref. 22), the influx is regulated by the concentration of ER Ca²⁺, as it should be for SOCE. I guess this is the case here too. It is worth reminding it. It would also be interested in seeing somewhere the evolution of ER Ca²⁺.

(3) About Panel 6D. How is Tg simulated? What does "maximal store depletion" mean? In fact, I am surprised to see that the effect of Tg application much differs depending on the Orai isoforms that are expressed. This is also the case in experiments, but I do not think that it is discussed anywhere. Moreover, Panel 6D does not seem to be in agreement with Fig. 1w in this respect.

Minor:

*On p11, line 362, it is written that there are some model results in the methods section. I did not see any.

*In Equation (11), d11 should be replaced by d12.

Not about modelling. How do the authors relate their observations with CDI? Is it assumed that CDI is very fast and that it explains why Orai1 mediates the most important Ca²⁺ influx? In the first paragraph of the discussion, the authors write that because Orai2 and 3 have greater CDI, they are tailored to mediate responses to low/mid agonist doses. Why? One could argue the opposite, i.e. that to sustain a response at low/mid agonist doses, the cell needs more Ca²⁺ to enter the cell (and reinforce the low dose of IP3).

Response to reviews:

“*The ORAI channel trio underlies the diversity of Ca²⁺ signaling events*” By Yoast et al.

We are very grateful to all three reviewers for their thorough reading of our manuscript and for their constructive criticisms. The reviewers have pointed out a significant number of important issues that we have failed to clarify in the first version. In this revision, we have provided clarifications, statistics and additional data requested by the reviewers. These data are the subject of two new tables (**Supplementary table 4 and 5**) and four additional Figures (**Supplementary Fig. 9, 10, 11 and 13**). All modifications to the text are highlighted in yellow and below is our response point by point to the comments by the three reviewers:

Reviewer 1 (Remarks to the Author):

The manuscript is arranged in a nice fashion and all experiments are presented with a clear train of thought. A wide spectrum of different methods including cell culture, molecular cloning, CRISPR/Cas9, quantitative PCR, Ca²⁺ imaging, FRET microscopy, biochemistry, electrophysiology and mathematical modeling enables the construction of a native model of a heteromultimeric CRAC channel complex composed of all three ORAI protein isoforms. The authors reveal a specific correlation between ORAI2/ORAI3 and cellular Ca²⁺ oscillations. While these do not depend on the presence of ORAI1, the latter is indispensable for robust SOCE and sustained Ca²⁺ plateau signals. ORAI2/ORAI3 additionally heteromerize with and negatively regulate ORAI1 by strengthening Ca²⁺-dependent inactivation (CDI). Finally, the nuclear factor of activated T-cells (NFAT) family requires ORAI1 for activation with ORAI2/ORAI3 alone not being sufficient. However, NFAT isoforms require varying activation signals mainly differing in the amount of Ca²⁺ that is required. Through their negative regulation of ORAI1, ORAI2/ORAI3 enable the differential activation of NFAT isoforms in response to distinct agonists and cytosolic Ca²⁺ concentrations. With this study, the authors thus provide another key step towards elucidating the molecular choreography of the native CRAC channel complex and the diverse Ca²⁺ signaling events supported by it.

We sincerely thank reviewer 1 for their positive assessment of our work.

The implication that both ORAI2 as well as ORAI3 heteromerize preferentially rather than forming homomeric assemblies needs to be proven experimentally.

We thank reviewer 1 for this important comment. We have thoroughly considered this idea and believe proving this statement experimentally under native conditions is not possible with current methods. Even if we undertake the laborious and uncertain task of over-expressing tagged proteins, purifying them and titrating binding, binding in biological membranes is likely different than that in solution. We could attempt similar overexpression experiments of tagged proteins coupled with co-immunoprecipitations, *in cellulo* FRET or other methods to determine binding. However, even if this strategy is successful and the results obtained are clear-cut, it might not necessarily reflect binding under native conditions. We do not refute the possibility that homomeric channels consisting of ORAI2 or ORAI3 also exist, since many investigations

including our own showed that ORAI can assemble as both homo- and heter-multimers. The mathematical model and the rescue experiments in ORAI-TKO cells suggest that ORAI1 is expressed more than ORAI2 and ORAI3. Based on our data, the WT HEK293 situation does not simply reflect the sum of various ORAI KO cells, arguing that the three ORAI hetermultimerize. The deletion of either ORAI2 or ORAI3 from the native system significantly reshapes the cytosolic Ca^{2+} signal providing evidence that ORAI2 and ORAI3 are members of the native CRAC channel.

The stronger interaction of ORAI2/3 with STIM1 compared to ORAI1 is suggested to result in Ca^{2+} entry at low- and mid-range agonist concentrations which only slightly deplete Ca^{2+} from the ER store.

We agree with this statement provided by reviewer 1. We assume that reviewer 1 is asking why despite the increased basal interaction of ORAI2 and ORAI3 with STIM1, cells lacking ORAI1 do not respond to lower concentrations of agonist? This apparent discrepancy is explained by the native CRAC channel being a heteromultimer of ORAI isoforms. In this heteromeric context, ORAI2 and ORAI3 are negative regulators of ORAI1. Because ORAI2 and ORAI3 have low channel activity compared to ORAI1, when the native CRAC channels is deprived of ORAI1 they are not as effective in replenishing ER Ca^{2+} stores.

To experimentally address these issues under native conditions would require the generation of different combinations of fluorescent protein-tagged knock-ins of STIM and ORAI isoforms and extensive studies imaging their movement and interactions in response to various concentrations of agonist. These are laborious experiments and we hope reviewer 1 agrees that they are beyond the scope of this study.

The impact of CDI would be interesting to see with ORAI channels devoid of it.

We thank reviewer 1 for this interesting point regarding CDI of ORAI channels. In light of reviewer 1 comment we have pondered means to gain some insights into the contribution of CDI of different ORAI isoforms in shaping native agonist-induced Ca^{2+} signals. The residues that affect CDI have been described only for the case of ORAI1^{1,2}. To adequately address this question under native conditions of expression, we would have to determine the mutations required to alter CDI in ORAI2 and ORAI3 and subsequently generate knock-in of these mutations. Because of the small nature of native CRAC currents, studying CDI under these native levels of ORAI expression is extremely difficult if not impossible. These experiments, which are laborious and uncertain, do not impact on the conclusions of the current paper and we hope reviewer 1 agrees that they are beyond the scope of this study.

Low agonist concentrations have been recently shown to mediate STIM1 activation via STIM2. Could this play here a role as well, as STIM2 is assumedly not knocked-out?

We thank reviewer 1 for this keen observation. Using a similar strategy of double ORAI KO combined with single STIM KO (cells with one native STIM and one native ORAI isoform) we have established a collaborative role for STIM2 and STIM1 in regulating the Ca^{2+} oscillatory frequency, which are consistent with the work published by Dr. Indu Ambudkar. These extensive

data, which are the subject of another manuscript currently in preparation, do not impact on the conclusions of the current paper. We believe these extensive STIM1/2 data are better addressed in a separate study as they will unnecessarily increase the complexity of the current paper, which already has ample data.

The authors suggest that native expression levels will be achieved by using the weak tk promoter instead of the strong CMV-promoter (line258), however, there is no experimental proof of that.

We apologize for this omission. We have previously compared side by side using Western blots and fluorescence imaging the expression of ORAI1 driven by either the tk or the CMV promoters (please see **Supplemental Figure 1a-b** from Zhang et al³) and show that the tk promoter achieves near-native levels of ORAI1 expression. We now have updated the text to reflect this clarification (please see highlighted text in the results).

CRISPR/Cas9 method used: The authors state that they used two different methods for knock-out construction. First, it is not clear to me which cell clones (that were used within the manuscript) derived from which method.

We apologize for this. We have addressed the lack of clarity noted by both reviewers 1 and 2 surrounding the CRISPR clones. We have documented all genomic changes induced by our CRISPR/Cas9 system on an allele by allele basis. Furthermore, we recorded which clones were generated with a single or two gRNAs. To enhance clarity, we have decided to include all the necessary information related to CRISPR/Cas9 KO clones in a new table (now **Supplementary Table 4**).

In Supplementary Figure 1a isoform expression in ORAI1 knock-out cells is shown. The image is not explained in detail and one can just estimate which bands can be seen on this image. Should the prominent band below 25 kDa correspond to ORAI1? If so, the authors need to state why there is a difference between observed sizes on western blot and the expected, calculated size of endogenous ORAI1 proteins of ~33kDa. Moreover, there are two additional bands visible right below 37kDa – there is no explanation given in the manuscript what these bands represent.

And

Figure 2o: whole blots have to be shown and the question arises how the size of ORAI1 shown here (37kDa) can be correlated to the different sizes shown in Supplementary Figure 1a.

We are grateful to reviewer 1 for noticing this issue. We apologize for not clarifying the differences between the two ORAI1 blots, which use two different protocols. The ORAI1 antibody generally resolves a broad band that ranges from ~25-40kDa. This is due to the presence of two translational variants, ORAI1 α (33kDa) and ORAI1 β (25kDa) that are both heavily glycosylated and recognized by the antibody. To clearly resolve ORAI1, and verify genetic knockout, we deglycosylated the protein lysates using the enzyme PNGase F (please also see Zhang et al³). Using this protocol allowed us to resolve both translational variants of ORAI1 and clearly verify ORAI1 knockout at the protein level. We have updated the figure legend to clearly outline the rationale for using PNGase F. Labels have also been added to **Supplementary Figure 1a** to distinguish ORAI1 α from ORAI1 β clearly. The protein lysates for figure 2o were

not treated with PNGase F. Because of this, glycosylated ORAI1 α and ORAI1 β manifest as a broad smear rather than the sharp two bands seen in **Supplementary Figure 1a**.

gRNA is targeting ORAI1 at ~amino acid 145. Evidence is given by literature that the DSB usually occurs 100bp upstream or downstream of the gRNA binding site. Still, designing the gRNA this way, there will be ~110 amino acids expressed of the native ORAI1 protein. As the ORAI1 antibody used for experiments binds to a peptide at the very end of the ORAI1 protein (288-301) the authors can not exclude that there is still a functional part of ORAI1 left.

In conjunction with the previous point, some of the cell lines used in this paper (ORAI3-SKO Clone #25, ORAI2,3-DKO Clones #9 and #22) also allow the translation of small residual ORAI3 protein fragments that could potentially interfere with other ORAI isoforms or with ORAI3 artificially introduced by transfection.

We thank Reviewer 1 for this insightful and valid question regarding our multiple CRISPR/Cas9 cell lines. Indeed, the CRISPR/Cas9 system we used generates a double-strand break at the site of gRNA homology. In all ORAI1 clones, there was a frameshift before sequences encoding transmembrane regions 3 and 4. In this case, if the channel is translated into a shorter protein, it is nonfunctional. We have validated this lack of function using patch-clamp electrophysiology (**Figure 5h**) and Fura2 Ca²⁺ imaging (**Figure 1r**). However, as reviewer 1 alluded to, the expression of a shorter ORAI protein, even if it is not functional as a channel, might act as dominant negative in cells. Although we don't have an antibody that can detect the potential existence of shorter N-terminal portions of ORAI, rescue experiments of ORAI isoforms in ORAI-TKO cells (**Figure 3**) rule out potential dominant negative effects of a shorter ORAI1 protein potentially expressed in the KO cells.

Supplementary Table 2: ORAI1 gRNA does not match the sequence given in the methods section. It is extended by CACG-, a sequence that does not bind ORAI1.

Thank you for noticing this discrepancy, we have corrected this typo and included all gRNA sequences in **Supplementary Table 4**.

Supplementary Figure 2b/c: In Supplementary Figure 2b and 2c, the PCR results of genomic DNA are intended to show large genomic deletions of ORAI2 to validate gene knockout. However, several issues with these figures remain to be clarified: (1) it is unclear whether the WT and KO boxes belong to the same gel or show different gels, (2) WT seems to indicate ORAI1-SKO cells, probably confusing the reader, (3) it is not explained why more than one band appears for clones #19 and #22, and (4) along with the issue raised in (1), the presence of size-marker labeling would make it easier to interpret the results.

Per reviewer 1 suggestion and to avoid any confusion, we have now included all relevant information in **Supplementary Table 4**, which explains the existence of two bands. Alleles 1 and 2 differ in their mutational profile. Allele 1 had a large deletion and insertion whereas allele 2 only had a large deletion. The two different mutations result in two knockout bands with a difference in length.

There are several typing-errors throughout the text (line 121, 132, 159, 392, ..)

Thank you, we have thoroughly re-read and edited the text for typographical issues. Please see the highlighted text throughout the revision.

Methods section – FRET microscopy: it is stated that a STIM1-mCherry plasmid was used for transfection – but this does not show up anywhere in the figures.

We thank reviewer 1 for noticing this important experimental detail and we apologize for failing to include an explanation. The STIM1-mCherry was included to ensure the proper activation of the overexpressed ORAI channels. Overexpression of ORAI1 alone (without STIM1) results in dominant negative effects. We chose a tagged STIM1 construct to make sure we are recordings from cells that indeed co-express STIM1 and two ORAI isoforms. Specifically, mCherry-STIM1 plasmid was chosen due to its excitation and emission spectra being outside of the FRET pair range. We have now clarified this point as highlighted in the text.

ORAI isoform heteromerization is mimicked by the use of concatenated dimers. In line 312/313 the authors claim that peak currents of dimer constructs are smaller due to the lower expression of these constructs. Yet, there are no experiments presented where expression levels and correct localization of dimer constructs are shown.

The ORAI concatenated homodimers have been previously published⁴, where we have shown that these constructs localize to the plasma membrane. In that study, the ORAI dimers were tagged with the much brighter tdTomato because precisely they generated lower fluorescence in cells compared to the ORAI monomer when tagged with CFP or YFP. Because ORAI dimers also generate smaller whole-cell CRAC currents, this is an indication of lower expression (please compare **Supplementary Fig 4c** with **Supplementary Fig 8c**). Nevertheless, we changed the text to reflect a less definitive tone by stating in page 9 “*Peak currents generated by expression of ORAI concatenated dimer constructs are smaller, likely reflecting a relatively lower expression of these constructs*”. We have now included images of the newly generated ORAI1-2, ORAI1-3, and ORAI2-3 heterodimers in **Supplemental Figure 13** that demonstrate the proper membrane localization of these constructs in ORAI-TKO cells as kindly requested by reviewer 1.

The authors make excessive use of subfigures, a number of which could easily be merged in some form (e.g. Figure 1 r-w). The figures would also benefit from the use of visual cues to indicate/group related subfigures. Finally, the references for significance testing should be better indicated in all cases, as simple asterisks (*) do not fully clarify that comparisons are only related to wild type cells (cf. comment to Figure 1x below).

Per reviewer 1 suggestion, we have combined multiple subfigures in **Figure 7** as this was feasible. There was no easy way to combine other subfigures without creating further confusion. After some experimentation with the figures and the related text, we concluded that inclusion of all subfigures is necessary to avoid confusing the reader. For example, the suggested combination of **Figures 1 r-w** results in a single graph that is difficult to follow and interpret. There are simply too many conditions to be displayed on a single set of axes to resolve the differences between all 6 cell lines.

Per reviewer 1 suggestion, we now have included multiple comparisons in the figures to substantiate claims within the text. Simple asterisks were only used when various comparisons became too numerous thus severely hampering the clarity of the graph (e.g. **Figure 1m**). The methods used to analyze all datasets are clearly outlined within the figure legends.

The number of cells used for the calculation of mean values and their corresponding SEM are not indicated in many applicable subfigures across the manuscript.

Thank you! We have now included the “n” value for all necessary scatterplots in the figure legends. This information can also be found in the source data file.

It is claimed that ORAI3-SKO cells show “increased SOCE in response to 2 μ M thapsigargin” and “increased oscillatory frequency triggered by 10 μ M carbachol” when compared to wild type cells, although I couldn’t see the underlying statistical analysis.

Both reviewers 1 and 2 noticed this discrepancy between the data and our text. The text now clearly states that the small increase is not statistically significant for ORAI3-SKO cells.

With reference to Figure 1x, the authors report that “ORAI2,3-DKO cells show an increase in SOCE in response to thapsigargin” when compared to wild type cells and that “this increase was significantly bigger than that of ORAI2-SKO and ORAI3-SKO cells” (l 175). However, all significance testing in Figure 1x relates to wild type cells (as is also indicated in the corresponding figure legend).

We thank both reviewers 1 and 2 for bringing up this point. We have performed the appropriate statistical test, a Kruskal-Wallis test with multiple comparisons, to compare ORAI2,3-DKO cells with ORAI2-SKO and ORAI3-SKO cells (**Fig 1x**). Our results concluded that ORAI2,3-DKO cells do indeed have more SOCE than either ORAI2-SKO or ORAI3-SKO cells alone. We have added the appropriate annotations to **Figure 1x** to indicate the inclusion of this test and have updated the figure legend to reflect the change.

It would be interesting to see respective ORAI localization images when expressed in ORAI-TKO cells.

Thank you! To show the localization of ORAI1, ORAI2, ORAI3, and the dimer constructs not previously published (i.e. ORAI1,2, ORAI1,3, and ORAI2,3 dimers), we have included **Supplementary Figure 13**. In this figure all constructs were expressed in ORAI-TKO cells.

Line 360: Mathematical model taking only “ORAI dimer formation” into account: I guess the meaning is that only two ORAI isoforms contribute to the hexameric ORAI channel.

The model is even more restrictive than stated by reviewer 1. The model assumes that only dimers can form. Thus, the ORAI channels are not hexamers; they consist of only two ORAI molecules. Without this assumption, the combinatorial complexities make the modelling much more difficult.

Reviewer #2 (Remarks to the Author):

In the current study, Yoast et al. perform a comprehensive analysis of the contribution of each of the three ORAI proteins in HEK cells for calcium homeostasis following activation of agonist/receptor-operated and store-operated Ca²⁺ entry pathways. They generate and analyse seven different genotypes after generation single, double- and triple-compound ORAI1 knockout alleles in HEK cells covering all possible combinations of ORAI subtype deletions to study the distinctive role of individual ORAI proteins for calcium entry and/or oscillations generated by different strengths of receptor stimulation by comparison with wildtype HEK cells. The authors study heteromultimerization in this cellular system using FRET analysis upon expression of fluorescently tagged ORAI1 proteins or expression of individual ORAI proteins with established pore mutations. The readout for a cellular function that can be attributed to the distinct calcium signaling pattern conveyed by individual ORAI proteins is nuclear translocation of transcription factor NFAT1 and NFAT4. Using this approach, the authors elaborate an essential role of ORAI1 as the key ORAI subtype mediating store-operated calcium entry as determinant of NFAT1 and NFAT4 translocation to the nucleus. There are several issues that need to be addressed (see below), but overall this is a very carefully performed study analyzing systematically and comprehensively the contribution of all ORAI isoforms in a given cell system under native expression levels which has not been demonstrated before in the literature. Overall the results are convincing and support the conclusions.

We thank reviewer 2 for their positive assessment of our manuscript.

1. The generation of the individual and compound ORAI knockout cell lines seems to be carefully performed, but the documentation of the molecular characterization needs to be documented in more detail in the supplementary figures where space is not limited. The authors should provide the sequence (deletions, insertions) for all individual alleles for ORAI1, ORAI2 and ORAI3 single knockouts as well as the double and triple knockout cell clones. The analysis of compensatory up- and down-regulation of STIM and IP3 receptor proteins is well performed. However, up- or down-regulation of ORAI2 and ORAI3, respectively, should be addressed in all knockout clones by qPCR. This has only been partially performed in supplementary figure 2, e.g. expression of ORAI3 should be shown in ORAI1&2 double knockout clones (# 18 and 19), ORAI2 expression in ORAI1/3 double knockout (clone #48, clone #3) and similarly in ORAI2 SKO and ORAI3 SKO clones. Also, supplementary figure 2b and c are difficult to understand. They should be correlated with the documentation of the gene edited sequences of the individual clones in more detail. As no specific antibodies are available for ORAI2 and ORAI3, respectively, the authors should compare the expression levels of ORAI1, 2 and 3 transcripts in their cell system, i.e. HEK cells (may be they have RNA seq data for this).

Both reviewers 1 and 2 had similar questions regarding the process used to validate the multiple CRISPR cell lines used throughout the manuscript. We have now clearly annotated the changes to the genomic DNA or each CRISPR clone allele by allele in **Supplementary Table 4**. This table also includes the exact gRNA sequences used to generate each cell line.

2. The authors should define more precisely what qualifies cellular response into the category “plateau cells”. The authors only state that the intracellular calcium level should be above

baseline for longer than five minutes. What is above baseline? more than 5% or 2% or what is the cut off?

We thank reviewer 2 for this suggestion. We have now included a new figure to illustrate the three established cytosolic Ca^{2+} phenotypes (No-Response, Oscillating, and Plateau) (**Supplementary Figure 11**). We have also included the line “The third group (plateau cells) have a cytosolic Ca^{2+} signal that is $\geq 25\%$ of the initial peak for at least 5min post-stimulation...”

5. On page 8, line 269 the authors state that TK promotor driven ORAI3 expression leads to significant SOCE. In figure 3a (green average trace) the Ca^{2+} entry seems not to be significantly larger than in the ORAI1 TKO cells. Please comment and provide quantitative comparison.

We have confirmed that cells transfected with tk-ORAI3 have significantly more SOCE than untransfected ORAI-TKO cells. Using the Kruskal-Wallis test with multiple comparisons a p-value of 0.0002 was observed when comparing tk-ORAI3 transfected cells to ORAI-TKO cells.

6. In figure 5 the authors study ORAI1 pore mutant in ORAI1-SKO cells (A, B). The authors then proceed and study ORAI2 and ORAI3 pore mutants (C, D), respectively again in ORAI1-SKO cells. Shouldn't the ORAI2 pore mutant be studied in ORAI2-SKO and ORAI3 pore mutant in ORAI3-SKO to study functional interaction with the other corresponding ORAI isotypes?

The rationale for experiments in figure 5 utilizing ORAI1-SKO cells is to block the residual SOCE mediated by ORAI2 and ORAI3 within the ORAI1-SKO cells and observe the effects on Ca^{2+} oscillations. We hypothesized that introduction of a dominant-negative ORAI constructs and their heteromultimerization with endogenous ORAI2 and ORAI3 subunits would fully inhibit Ca^{2+} oscillations. Interestingly, we found that introduction of dominant-negative ORAI in ORAI1-SKO cells only partially inhibited Ca^{2+} oscillations, suggesting that minuscule channel activity of the dominant-negative ORAIs is sufficient to partially support Ca^{2+} oscillations.

7. The authors make use in figure 5g-m of 2-APB which has different action on ORAI1, ORAI2 and ORAI3. Interestingly, the small but remaining current with a typical I-V curve of a CRAC channel is enhanced by 2-APB in ORAI1-SKO cells. Are ORAI2- and ORAI3-mediated currents sensitive to other typical CRAC channel blockers such as GSK7975A?

Reviewer 2 is correct that the current is enhanced when ORAI1-SKO cells are treated with 2-APB, reflecting the store-independent activation of ORAI3 by 2-APB⁵. To address reviewer 2 second question, we have performed experiments using GSK7975A as requested by reviewer 2 (**Supplementary Figure 9**). The results determined that ORAI1, ORAI2, and ORAI3 are all inhibited by GSK7975A.

8. Page 10 line 354: line 352 the authors state that 2-APB potentiates currents in cells lacking ORAI2 (figure 5i). This statement is not supported by the data shown.

When we consider each ORAI-SKO and each ORAI-DKO cells, CRAC currents are only observed in cells expressing native ORAI1 and this current is blocked by 2-APB. In single and double ORAI-KO clones, 2-APB-mediated CRAC current potentiation is detected only in cells that have preserved ORAI3 expression but lack expression of either ORAI1 (**Figure 5h**), or

ORAI2 (**Figure 5i**) or both (**Figure 5m**). For the case of ORAI2-SKO cells specifically mentioned in reviewer 2's comment (**Figure 5i**), native CRAC currents (mediated by ORAI1) remain unaffected after addition of 50 μ M 2-APB. The easiest interpretation of these results is the inhibition of native ORAI1 by 2-APB and concomitant activation of native ORAI3.

9. In figure 7 the authors show convincingly that nuclear translocation of NFAT1 and NFAT4 is differently regulated upon deletion of ORAI1 on the one hand and ORAI2/3 inactivation on the other hand. These experiments correlate very well with the effect of the individual ORAI deletion on the extent of an amplitude of SOCE. ORAI2 and 3 are only modulating but not responsible for the amplitude of store-operated calcium entry (as opposed to ORAI1) but are key regulators of calcium oscillations, particularly at the carbachol concentrations studied (10mM); thus, the authors should aim to analyze the activation of transcription factors that do not depend on a plateau calcium elevation but are regulated by the frequency of calcium oscillations. NFkB has been reported already many years ago as such calcium dependent transcription factor (Dolmetsch et al., Nature 386: 855-8, 1997) and should be studied in the ORAI Ko cells (ORAI1 vs ORAI2/3) or other transcriptional responses for which the authors could show dependence on the Ca²⁺ oscillation frequency.

We thank reviewer 2 for this suggestion; we had considered this idea around the same time we performed NFAT nuclear translocation experiments. Thus, we analyzed the transcription factor cFOS and its potential selective activation by ORAI1 in response to low agonist stimulation, as suggested in the literature. However, despite multiple attempts, the results of these experiments were inconclusive. We agree with reviewer 2 that NFkB activation might depend on a single ORAI isoform. This is a logical next step that is worth analyzing but would hope that reviewer 2 agrees that this question is better considered under a separate future manuscript.

Reviewer #3 (Remarks to the Author):

In this manuscript, the authors investigate the roles of ORAI2 and ORAI3 in Ca²⁺ entry in physiological conditions, i.e. when cells are stimulated by moderate doses of agonists. The amount of results is impressive, and the experiments are very clear. The manuscript represents a significant contribution to the understanding of ORAI-mediated Ca²⁺ influx and to Ca²⁺ signaling in general.

We thank reviewer 3 for their kind assessment of their work.

(1) How were the d_{ij} values determined? The authors write that it was done "by comparing the model output qualitatively to the experimental results". Intuitively, I would have fixed the ratios of the d_{ij} by looking at the experiments shown in Fig. 1, panels r to w. It is obviously not what has been done. Why? The qualitative procedure should be explained.

Estimation of the d_{ij} relies on the fact that the model responses can be characterized by the maximal influx current (J_{in}). In other words, J_{in} can be used as a bifurcation parameter (instead of, say, the level of agonist stimulation, as is more usual) and bifurcation diagrams can be constructed as usual. Thus, one can construct a mapping between J_{in} and model behavior (in particular the shape and period of oscillations). Consequently, if a certain model behavior is

desired, this can be done by consulting the bifurcation diagram and selecting the appropriate value for J_{in} , and this is the approach we took. For each observed oscillatory pattern we simply chose the value of J_{in} that gave the desired behavior.

Our approach means that we are sure of the (approximately) correct oscillatory responses but unsure of the behavior in the thapsigargin experiments, which must be determined without additional constraints on the model.

Using the experiments in **Figure 1** would certainly be another way of estimating the values for d_{ij} . In that scenario we would have ensured that the thapsigargin (Tg) experiments were reproduced exactly, while leaving the oscillations to fall as they may.

Either approach is valid.

(2) In the PNAS paper (ref. 22), the influx is regulated by the concentration of ER Ca^{2+} , as it should be for SOCE. I guess this is the case here too. It is worth reminding it. It would also be interested in seeing somewhere the evolution of ER Ca^{2+} .

We now state explicitly in the paper that influx is a decreasing sigmoidal function of ER Ca^{2+} concentration, please see highlighted text. Furthermore, we show some examples of ER Ca^{2+} concentration (**Supplementary Fig. 10**).

(3) About Panel 6D. How is Tg simulated? What does “maximal store depletion” mean? In fact, I am surprised to see that the effect of Tg application much differs depending on the ORAI isoforms that are expressed. This is also the case in experiments, but I do not think that it is discussed anywhere. Moreover, Panel 6D does not seem to be in agreement with Fig. 1w in this respect.

In the model, thapsigargin (Tg) is simulated by decreasing the SERCA pump flux to zero. This is now mentioned explicitly within the text.

Use of the word "maximal" is incorrect (at least as far as the model is concerned). We have now corrected the sentence to read "depletion of the store to less than 20% of its resting state".

The differing effects of Tg application, followed by the reintroduction of external Ca^{2+} , are a result of the interactions between ORAIs. Since ORAI2 and ORAI3 are assumed to inhibit ORAI1, as suggested by the experimental results, the ORAI2,3-DKO case gives a much larger response than WT, as ORAI1 is relieved from inhibition. This is now briefly emphasized in the paper.

Figure 6D is inconsistent with **Figure 1w** in that, in the model, the first rise in Ca^{2+} (upon addition of Tg) is different for the WT and ORAI1,2-DKO cases, while in the experiment it is the same. The model is in clear disagreement with the experiments regarding this ER Ca^{2+} release phase. We know why this happens in the model; in the ORAI1,2-DKO case there is a decreased steady-state influx of Ca^{2+} from outside the cell, and this necessarily results in a lower ER Ca^{2+} concentration, and thus a smaller response upon the introduction of Tg. There is no avoiding this in the model. Indeed, it is a puzzle to us why the experiments do not show the same. In fact, in studies assessing STIM and ORAI-KO cells from animals and studies on

cultured cells lacking both STIM1 and STIM2⁶ or cells lacking all three ORAIs (the current study), the ER Ca²⁺ content appears normal. It is possible that the ER Ca²⁺ concentration, at steady state, is independent of cytosolic Ca²⁺ rise via some unknown mechanism. Alternatively, it is possible that at steady state, an extremely small ORAI-independent Ca²⁺ influx pathway could refill the stores in a matter of days.

Minor: On p11, line 362, it is written that there are some model results in the methods section. I did not see any.

We were thinking of **Figure 6** as being in the Methods section, but we see now that our wording was confusing. We have changed this sentence to say that model details are in the methods section and now appropriately reference **Figure 6** and **Supplementary Figure 10**.

In Equation (11), d11 should be replaced by d12.

This has now been corrected. We thank the reviewer for noticing this error.

Not about modelling. How do the authors relate their observations with CDI? Is it assumed that CDI is very fast and that it explains why ORAI1 mediates the most important Ca²⁺ influx? In the first paragraph of the discussion, the authors write that because ORAI2 and 3 have greater CDI, they are tailored to mediate responses to low/mid agonist doses. Why? One could argue the opposite, i.e. that to sustain a response at low/mid agonist doses, the cell needs more Ca²⁺ to enter the cell (and reinforce the low dose of IP3).

We Thank reviewer 3 for this question. Our thinking considers the native CRAC channel as a heterohexamer of different ORAIs and is as follows: At low concentrations of agonist, the Ca²⁺ responses are small and these responses increase with increasing concentrations of agonist, thus effectively tailoring the Ca²⁺ responses to the strength of agonist stimulation. At low agonist concentrations, the presence of ORAI2 and/or ORAI3 within a native heterohexameric CRAC channel would dampen the Ca²⁺ responses to match the low level of agonist stimulation.

- 1 Srikanth, S., Jung, H. J., Ribalet, B. & Gwack, Y. The intracellular loop of Orai1 plays a central role in fast inactivation of Ca²⁺ release-activated Ca²⁺ channels. *The Journal of biological chemistry* **285**, 5066-5075, doi:10.1074/jbc.M109.072736 (2010).
- 2 Mullins, F. M., Yen, M. & Lewis, R. S. Orai1 pore residues control CRAC channel inactivation independently of calmodulin. *The Journal of General Physiology* **147**, 137-152, doi:10.1085/jgp.201511437 (2016).
- 3 Zhang, X. *et al.* A calcium/cAMP signaling loop at the ORAI1 mouth drives channel inactivation to shape NFAT induction. *Nature Communications* **10**, 1971, doi:10.1038/s41467-019-09593-0 (2019).
- 4 Cai, X. *et al.* The Orai1 Store-operated Calcium Channel Functions as a Hexamer. *The Journal of biological chemistry* **291**, 25764-25775, doi:10.1074/jbc.M116.758813 (2016).
- 5 Schindl, R. *et al.* 2-aminoethoxydiphenyl borate alters selectivity of Orai3 channels by increasing their pore size. *J Biol Chem* **283**, 20261-20267, doi:10.1074/jbc.M803101200 (2008).

- 6 Emrich, S. M. *et al.* Cross-talk between N-terminal and C-terminal domains in stromal interaction molecule 2 (STIM2) determines enhanced STIM2 sensitivity. *J Biol Chem* **294**, 6318-6332, doi:10.1074/jbc.RA118.006801 (2019).

REVIEWERS' COMMENTS:

Reviewer #1 (Remarks to the Author):

No further comments.

Reviewer #2 (Remarks to the Author):

In principle, the authors have sufficiently addressed all my concerns.

One remaining question is regarding Suppl. Fig2: the qPCR analysis in Orai3 SKO shows strongly reduced levels of O3 transcripts in clone #9, but not in clone #25.

Clone #9 exhibits a 332 nt deletion on both alleles, clone #25 a deletion of one nucleotide on one allele and an insertion of 183 nt in the other allele, which may explain the qPCR results?

In the qPCR analysis of Orai 2, 3 DKO clones, which are derived from Orai3-SKO parental clone #25, strongly reduced levels of O3 transcripts were then found in Orai 2, 3 DKO clone #9 but no significant change in Orai 2, 3 DKO clone #22.

What is the explanation for this finding? I have difficulties to understand this finding as the Orai3 gene was not targeted anymore in this second genome editing step, but only the Orai2 gene in both Orai 2, 3 DKO clone #22 and clone #9.

Reviewer #3 (Remarks to the Author):

The authors have answered my questions/remarks and modified the manuscript accordingly.

Reviewer #1 (Remarks to the Author):

No further comments.

Reviewer #2 (Remarks to the Author):

In principle, the authors have sufficiently addressed all my concerns.

One remaining question is regarding Suppl. Fig2: the qPCR analysis in Orai3 SKO shows strongly reduced levels of O3 transcripts in clone #9, but not in clone #25.

Clone #9 exhibits a 332 nt deletion on both alleles, clone #25 a deletion of one nucleotide on one allele and an insertion of 183 nt in the other allele, which may explain the qPCR results?

In the qPCR analysis of Orai 2, 3 DKO clones, which are derived from Orai3-SKO parental clone #25, strongly reduced levels of O3 transcripts were then found in Orai 2, 3 DKO clone #9 but no significant change in Orai 2, 3 DKO clone #22.

What is the explanation for this finding? I have difficulties to understand this finding as the Orai3 gene was not targeted anymore in this second genome editing step, but only the Orai2 gene in both Orai 2, 3 DKO clone #22 and clone #9.

Response: We are grateful to reviewer 2 for noticing this discrepancy. This is simply an unfortunate mix-up with other clones bearing the same number and we apologize for this. The two ORAI2,3-DKO clones (#9 and #22) both indeed show normal levels of ORAI3 mRNA comparable to WT control HEK293 cells since the parental line is ORAI3-SKO clone #25. We have updated Fig S2 accordingly with the correct data consisting of control WT cells and the two ORAI2,3-DKO clones run side by side and showing levels of ORAI3 mRNA comparable to those of WT control cells (see Fig. S2 attached).

Reviewer #3 (Remarks to the Author):

The authors have answered my questions/remarks and modified the manuscript accordingly.